# BILBO: BILevel Bayesian Optimization

**Ruth Wan Theng Chew** [1]   **Quoc Phong Nguyen** [2]   **Bryan Kian Hsiang Low** [3]

## Abstract

Bilevel optimization is characterized by a two-level optimization structure, where the upper-level problem is constrained by optimal lower-level solutions, and such structures are prevalent in real-world problems. The constraint by optimal lower-level solutions poses significant challenges, especially in noisy, constrained, and derivative-free settings, as repeating lower-level optimizations is sample inefficient and predicted lower-level solutions may be suboptimal. We present BILevel Bayesian Optimization (BILBO), a novel Bayesian optimization algorithm for general bilevel problems with blackbox functions, which optimizes both upper- and lower-level problems simultaneously, without the repeated lower-level optimization required by existing methods. BILBO samples from confidence-bounds based trusted sets, which bounds the suboptimality on the lower level. Moreover, BILBO selects only one function query per iteration, where the function query selection strategy incorporates the uncertainty of estimated lower-level solutions and includes a conditional reassignment of the query to encourage exploration of the lower-level objective. The performance of BILBO is theoretically guaranteed with a sublinear regret bound for commonly used kernels and is empirically evaluated on several synthetic and real-world problems.

## 1. Introduction

Many real-world problems involve hierarchical decision-making with two levels of optimization. Decisions made at the upper level affect lower-level optimization, while optimal lower-level solutions constrain decisions at the upper level. Bilevel optimization effectively models such hierar-

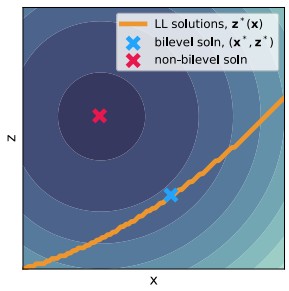 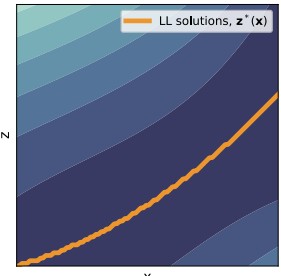

(a) Upper-level objective      (b) Lower-level objective

Figure 1: Example of bilevel optimization with upper-level variable $\mathbf{x}$ and lower-level variable $\mathbf{z}$. The bilevel solution (blue cross) is constrained by lower-level (LL) solutions (yellow line) and differs from the non-bilevel solution (red cross).

chical structures, enabling analysis of these interdependent problems. A simple example of bilevel optimization is in Figure 1. Applications of bilevel optimization range from machine learning (e.g., hyperparameter optimization, meta-learning) to economic problems (e.g., pricing strategies, toll setting) (Beck & Schmidt, 2021). In energy management, energy providers determine optimal pricing strategies for electricity (upper level) while consumers optimize their electricity demands based on the pricing (lower level). Similarly, in investment, brokers set fees on different asset classes to maximize their revenues (upper level), while investors optimize their portfolios for expected returns and risk (lower level). Bilevel optimization has been applied in both cases (Shu et al., 2018; Leal et al., 2020), typically using a nested framework with linear solvers at the lower level. This approach may limit practical effectiveness but it is due to the inherent complexity of bilevel optimization. Even with only linear constraints and objective functions, the set of feasible solutions can be non-convex and non-continuous (Kleinert et al., 2021). Lower-level solutions that are $\epsilon$-feasible w.r.t. non-linear constraints may also lead to a solution arbitrarily far from the bilevel solution (Beck et al., 2023).

Classical approaches (Bard & Falk, 1982; Bard & Moore, 1990) rely on simplifying linear or convex assumptions. More recently, meta-modeling methods employ surrogate models, such as BLEAQ (Sinha et al., 2013) which uses

[1]Institute of Data Science, National University of Singapore, Singapore [2]Amazon, Australia [3]Department of Computer Science, National University of Singapore, Singapore. Correspondence to: Ruth Wan Theng Chew <ruthchew@nus.edu.sg>.

*Proceedings of the 42$^{nd}$ International Conference on Machine Learning*, Vancouver, Canada. PMLR 267, 2025. Copyright 2025 by the author(s).

quadratic approximations to map upper-level points to lower-level solutions. We focus on meta-modeling methods over gradient-based ones to tackle general bilevel problems which may contain noisy observations, derivative-free functions at both levels, constraints and discrete variables.

Bayesian optimization (BO), a popular meta-modeling method, has been employed in a nested framework for bilevel optimization (Kieffer et al., 2017). The nested framework optimizes the upper level via BO while separately optimizing the lower level at each upper-level query point. However, these repeated lower-level optimizations are sample inefficient and often assume the presence of gradients.

**Contributions.** We propose BILevel Bayesian Optimization (BILBO), a novel BO algorithm for general bilevel problems with blackbox functions and constraints, where both levels are optimized simultaneously unlike existing nested bilevel BO frameworks. This adds new complexities, such as the potential lower-level suboptimality of the query and the uncertainty of estimated lower-level solutions. To tackle these challenges, we introduce key components of BILBO:

- Confidence-bounds based trusted sets to reduce search space with theoretical guarantees. In particular, sampling our trusted set of lower-level solutions bounds lower-level objective regret, effectively bounding lower-level suboptimality for upper-level optimization.

- Function query strategy that includes the uncertainty of the estimated lower-level solution and a conditional re-assignment of the query point for more effective lower-level objective evaluation. This balances exploiting estimated lower-level solutions with exploring to refine these estimates. We show that our strategy leads to an instantaneous regret bound on the query.

We further show that BILBO has sublinear cumulative and simple regret bounds for commonly used kernels, and demonstrate its effectiveness empirically on several synthetic and real-world problems. The code is provided in https://github.com/chewwt/bilbo/, and a notation table is in Appendix A.

## 2. Related Work

**Bilevel Bayesian optimization.** Most existing bilevel Bayesian optimization methods, as mentioned in the introduction, apply BO only to the upper-level problem and rely on repeated lower-level optimizations at each upper-level query (Kieffer et al., 2017; Islam et al., 2018; Wang et al., 2021). Dogan & Prestwich (2023) introduced an acquisition function that conditions on lower-level solutions during upper-level optimization for information flow between both levels. However, these nested methods require

gradient information to estimate or refine lower-level solutions, making them unsuitable for derivative-free bilevel problems. Fu et al. (2024) provided theoretical guarantees for a nested bilevel framework with stochastic gradient descent at the lower level and BO at the upper level. The reliance on lower-level gradients means the analysis does not extend to general derivative-free bilevel problems as well. In contrast, our proposed method is capable of handling blackbox, derivative-free bilevel problems, and our theoretical analysis is applicable to these settings.

An exception to the nested framework is a very recent parallel work by Ekmekcioglu et al. (2024) on arXiv. However, it has no theoretical guarantees and cannot handle constraints, unlike our proposed algorithm.

**Constrained Bayesian optimization.** Several constrained BO algorithms have been proposed (Gardner et al., 2014; Gelbart et al., 2014; Hernández-Lobato et al., 2016; Eriksson & Poloczek, 2021; Takeno et al., 2022). In particular, Xu et al. (2023) and Nguyen et al. (2023) both introduced confidence-bound based optimistic estimations of the feasible set, with the former providing an infeasibility declaration scheme and the latter including a function query strategy for decoupled settings. These feasible set estimations guide sampling toward probable feasible points, improving sample efficiency. Compared to constrained optimization, bilevel optimization presents additional challenges due to the need to optimize a separate lower-level problem, where the optimal solutions are unknown and often estimated suboptimally.

**Comparison to other optimization problems.** While some optimization problems, such as robust optimization (Bogunovic et al., 2018; Kirschner et al., 2020) and composite objectives optimization (Astudillo & Frazier, 2019; Li & Scarlett, 2024), respectively involve an additional random variable and composite objective function, they remain single-level problems. In contrast, bilevel optimization involves a two-level hierarchical structure, where the upper-level is constrained by lower-level solutions, making it fundamentally different from these other settings.

## 3. Preliminaries

**Bilevel optimization.** Let $F$ and $f$, respectively be the upper- and lower-level black-box objective function, where $F, f : \mathcal{X} \times \mathcal{Z} \to \mathbb{R}$. Let $\mathcal{C}_{\mathrm{up}}, \mathcal{C}_{\mathrm{lo}}$, respectively be the set of upper- and lower-level black-box constraints where $c : \mathcal{X} \times \mathcal{Z} \to \mathbb{R}$, $\forall c \in \mathcal{C}_{\mathrm{up}} \cup \mathcal{C}_{\mathrm{lo}}$. The upper-level variable is denoted $\mathbf{x} \in \mathcal{X}$ and lower-level variable as $\mathbf{z} \in \mathcal{Z}$, where $\mathcal{X} \subset \mathbb{R}^{d_{\mathcal{X}}}$ and $\mathcal{Z} \subset \mathbb{R}^{d_{\mathcal{Z}}}$ are assumed to be finite. We consider a general bilevel optimization problem with constraints as

$$\max_{\mathbf{x} \in \mathcal{X}, \mathbf{z} \in \mathcal{P}(\mathbf{x})} F(\mathbf{x}, \mathbf{z}) \tag{3.1}$$

$$\text{s.t.} \quad C(\mathbf{x}, \mathbf{z}) \geq 0, \quad \forall C \in \mathcal{C}_{\mathrm{up}}, \tag{3.2}$$

where $\mathcal{P}(\mathbf{x})$ is the set of optimal lower-level solutions at the upper-level variable $\mathbf{x}$,

$$\mathcal{P}(\mathbf{x}) \triangleq \{\arg\max_{\mathbf{z} \in \mathcal{Z}} f(\mathbf{x}, \mathbf{z}) \mid c(\mathbf{x}, \mathbf{z}) \geq 0, \ \forall c \in \mathcal{C}_{\text{lo}}\}. \tag{3.3}$$

Let $(\mathbf{x}^*, \mathbf{z}^*)$ denote the optimal bilevel solution, and $(\mathbf{x}, \mathbf{z}^*(\mathbf{x}))$ denote the optimal lower-level solution w.r.t. $\mathbf{x}$, where $\mathbf{z}^* \triangleq \mathbf{z}^*(\mathbf{x}^*)$. We define the set of functions $\mathcal{F} \triangleq \{F, f\} \cup \mathcal{C}_{\text{up}} \cup \mathcal{C}_{\text{lo}}$. At each step $t \geq 1$, we select a query point $(\mathbf{x}_t, \mathbf{z}_t)$ and obtain noisy observations $y_h(\mathbf{x}_t, \mathbf{z}_t) \triangleq h(\mathbf{x}_t, \mathbf{z}_t) + \epsilon$ where $\epsilon \sim \mathcal{N}(0, \sigma_n^2), \forall h \in \mathcal{F}$. In a decoupled setting, a function query $h_t \in \mathcal{F}$ is selected and only $y_{h_t}(\mathbf{x}_t, \mathbf{z}_t)$ is observed. Observations are accumulated into $\mathcal{D}_{h_t,t} \triangleq \mathcal{D}_{h_t,t-1} \cup \{y_{h_t}(\mathbf{x}_t, \mathbf{z}_t)\}$ and $\mathcal{D}_{h,0}$ is the set of initial observations for function $h$. $(\mathbf{x}_t, \mathbf{z}_t)$ is commonly used to estimate the optimal solution $(\mathbf{x}^*, \mathbf{z}^*)$.

**Gaussian process.** Each function $h \in \mathcal{F}$ is modelled with a Gaussian process (GP). Let $\mathbf{xz}$ be a concatenation of $\mathbf{x}$ and $\mathbf{z}$. A $\mathcal{GP}_h(m_h(\mathbf{xz}), k_h(\mathbf{xz}, \mathbf{xz}'))$ is specified by a mean function $m_h(\mathbf{xz}) \triangleq \mathbb{E}[h(\mathbf{xz})]$ and covariance function $k_h(\mathbf{xz}, \mathbf{xz}') \triangleq \mathbb{E}[(h(\mathbf{xz}) - m_h(\mathbf{xz}))(h(\mathbf{xz}') - m_h(\mathbf{xz}'))]$ (Williams & Rasmussen, 2006). At iteration $t$, given query inputs $\mathbf{xz}_{:t-1}$ and noisy observations $\mathbf{y}_{h,t-1}$, the predictive distribution for $h$ is Gaussian: $h(\mathbf{xz}) \mid \mathbf{xz}_{:t-1}, \mathbf{y}_{h,t-1} \sim \mathcal{N}(\mu_{h,t-1}(\mathbf{xz}), \sigma_{h,t-1}^2(\mathbf{xz}))$. More Gaussian process details can be found in Appendix B.

**Bayesian optimization.** Given a prior distribution $P(h)$ and likelihood function $P(\mathcal{D}_{h_t,t}|h)$, the posterior distribution $P(h|\mathcal{D}_{h_t,t})$ can be calculated via Bayes' theorem. The prior distribution is often represented by a GP and the likelihood function is defined by the choice of GP kernel and hyperparameters. The posterior distribution is also a surrogate model for $h$. The point which maximizes an acquisition function $a_h(\mathbf{xz})$ is selected as the next point to evaluate function $h$ at (Brochu et al., 2010; Frazier, 2018; Garnett, 2023). Popular acquisition strategies include information-theoretic ones (Hennig & Schuler, 2012; Hernández-Lobato et al., 2014; Wang & Jegelka, 2017) and upper confidence bounds (Srinivas et al., 2010).

**Regrets.** Regret is defined as the loss in reward from not selecting the optimal point. Instantaneous regret $r_t$ measures this loss at time $t$, while cumulative regret $R_T \triangleq \sum_{t=1}^{T} r_t$ is the sum of instantaneous regrets over $T$ rounds. An algorithm is no-regret if $\lim_{T \to \infty} R_T/T = 0$ where cumulative regret is sublinear and convergence to the optimal point is guaranteed with a large enough $T$.

We define the instantaneous bilevel regret as

$$r_t \triangleq \max_{h \in \mathcal{F}} r_h(\mathbf{x}_t, \mathbf{z}_t), \tag{3.4}$$

where $\mathcal{F} \triangleq \{F, f\} \cup \mathcal{C}_{\text{up}} \cup \mathcal{C}_{\text{lo}}$. The upper- and lower-level instantaneous objective regrets are defined, respectively, as

$$r_F(\mathbf{x}_t, \mathbf{z}_t) \triangleq \max(0, F(\mathbf{x}^*, \mathbf{z}^*) - F(\mathbf{x}_t, \mathbf{z}_t)), \tag{3.5}$$

$$r_f(\mathbf{x}_t, \mathbf{z}_t) \triangleq f(\mathbf{x}_t, \mathbf{z}^*(\mathbf{x}_t)) - f(\mathbf{x}_t, \mathbf{z}_t), \tag{3.6}$$

and the instantaneous constraint regrets as

$$\forall c \in \mathcal{C}_{\text{up}} \cup \mathcal{C}_{\text{lo}}, \ \ r_c(\mathbf{x}_t, \mathbf{z}_t) \triangleq \max(0, -c(\mathbf{x}_t, \mathbf{z}_t)). \tag{3.7}$$

Note that $r_c$ is usually known as constraint violation and we have incorporated them into the bilevel regret for a more representative estimate of the optimality of a point. An algorithm that is no-regret will have all objective function regrets and constraint violations converge to 0. If the constraints and objective functions have different ranges, normalization can ensure a fairer representation of the overall regret.

## 4. Bilevel Bayesian optimization

Our method, called *BILevel Bayesian Optimization* (BILBO), optimizes both upper- and lower-level simultaneously, via sampling from trusted sets and conditional reassignment to explore the lower-level objective. We avoid the repeated lower-level optimization found in most existing bilevel BO literature, as points in our trusted sets are sufficiently good for upper-level optimization directly. This is supported theoretically as we show that points in the trusted sets have upper-bounded instantaneous regrets on constraints and the lower-level objective. We also introduce a function query strategy based on estimated regrets, where we incorporate the uncertainty of estimated lower-level solutions and a conditional reassignment for exploration of the lower-level objective, to address challenges posed by the optimal lower-level solutions constraint. This results in an instantaneous regret bound on the query point, and leads to a sublinear cumulative and simple regret bound for commonly used kernels. The key components are illustrated in Figure 2 and the pseudocode is in Algorithm 1.

The trusted sets are defined in Definitions 4.3 and 4.5, and Lemmas 4.4 and 4.6 provide instantaneous regret bounds on points in the trusted sets. Definition 4.7 defines the function query selection and Lemma 4.8 presents a instantaneous regret bound on the query. The cumulative regret bound is in Theorem 4.9 and the simple regret bound in Lemma 4.10.

First, we define the confidence bounds on which we use to build the trusted sets. Functions are bounded by the confidence bounds with high probability by Corollary 4.2.

**Definition 4.1** (Confidence bounds). For a function $h \in \mathcal{F}$ modelled by a Gaussian process (GP), $\forall \mathbf{x} \in \mathcal{X}, \mathbf{z} \in \mathcal{Z}$, and $t \geq 1$, let the upper and lower confidence bounds of $h(\mathbf{x}, \mathbf{z})$ be denoted, respectively, as

$$u_{h,t}(\mathbf{x}, \mathbf{z}) \triangleq \mu_{h,t-1}(\mathbf{x}, \mathbf{z}) + \beta_t^{1/2} \sigma_{h,t-1}(\mathbf{x}, \mathbf{z}), \tag{4.1}$$

$$l_{h,t}(\mathbf{x}, \mathbf{z}) \triangleq \mu_{h,t-1}(\mathbf{x}, \mathbf{z}) - \beta_t^{1/2} \sigma_{h,t-1}(\mathbf{x}, \mathbf{z}), \tag{4.2}$$

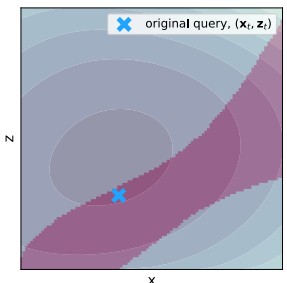

(a) Query point selection

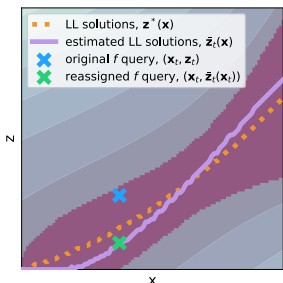

(b) Conditional reassignment

Figure 2: Key components of BILBO, where pink shaded areas represent trusted sets $\mathcal{S}_t^+ \cap \mathcal{P}_t^+$. (a) Query point selection within trusted sets following Equation 4.6. (b) Conditional reassignment following Equation 4.8, where lower-level query $\mathbf{z}_t$ may be reassigned to estimated lower-level solution $\bar{\mathbf{z}}_t(\mathbf{x}_t)$ if lower-level objective is selected for query.

where $\mu_{h,t-1}(\mathbf{x}, \mathbf{z})$ and $\sigma_{h,t-1}(\mathbf{x}, \mathbf{z})$ are the GP's posterior mean and standard deviation at $(\mathbf{x}, \mathbf{z})$, and $\beta_t \triangleq 2\log(|\mathcal{F}||\mathcal{X}||\mathcal{Z}|t^2\pi^2/(6\delta))$.

**Corollary 4.2.** *For some small $\delta > 0$, with probability at least $1 - \delta$, $\forall \mathbf{x} \in \mathcal{X}, \mathbf{z} \in \mathcal{Z}, h \in \mathcal{F}$, and $t \geq 1$,*

$$h(\mathbf{x}, \mathbf{z}) \in [l_{h,t}(\mathbf{x}, \mathbf{z}), u_{h,t}(\mathbf{x}, \mathbf{z})].$$

*This is derived from Lemma 5.1 of Srinivas et al. (2010) by applying union bound over $h \in \mathcal{F}$.*

Next, we introduce a trusted set of feasible solutions by approximating the unknown feasible regions using the upper confidence bound and show that points in this trusted set have constraint regret bounds.

**Definition 4.3** (Trusted set of feasible solutions). Let the trusted set of feasible solutions be defined as

$$\mathcal{S}_t^+ \triangleq \{(\mathbf{x}, \mathbf{z}) \in \mathcal{X} \times \mathcal{Z} \mid u_{c,t}(\mathbf{x}, \mathbf{z}) \geq 0 \ \forall c \in \mathcal{C}_{\text{up}} \cup \mathcal{C}_{\text{lo}}\},$$
(4.3)

where $\mathcal{C}_{\text{up}} \cup \mathcal{C}_{\text{lo}}$ is the set of all constraints, and the upper confidence bound $u_{c,t}$ is defined in Definition 4.1. For convenience, let $\mathcal{S}_t^+(\mathbf{x}) \triangleq \{\mathbf{z} \mid (\mathbf{x}, \mathbf{z}) \in \mathcal{S}_t^+\}$.

**Lemma 4.4.** *Let $\delta \in (0, 1)$, with probability at least $1 - \delta$, $\forall(\mathbf{x}, \mathbf{z}) \in \mathcal{S}_t^+, c \in \mathcal{C}_{up} \cup \mathcal{C}_{lo}$, the constraint regrets are upper bounded,*

$$r_c(\mathbf{x}, \mathbf{z}) \leq 2\beta_t^{1/2}\sigma_{c,t-1}(\mathbf{x}, \mathbf{z}).$$

The proof is provided in Appendix C.1. Sampling from $\mathcal{S}_t^+$ ensures that the instantaneous constraint regret of the chosen point is upper bounded, and highly infeasible points are outside the trusted set. An empty trusted feasible set would imply an infeasible bilevel problem, and our algorithm would make an infeasibility declaration.

**Algorithm 1** BILBO

---

**Require:** $\mathcal{X}, \mathcal{Z}, \{\mathcal{D}_{h,0}\}_{h \in \mathcal{F}}$
 1: Update GP posterior beliefs: $\{(\mu_{h,0}, \sigma_{h,0})\}_{h \in \mathcal{F}}$
 2: Update trusted sets $\mathcal{S}_t^+, \mathcal{P}_t^+$      // Defs. 4.3 and 4.5
 3: **for** $t \leftarrow 1$ to $T$ **do**
 4:    **if** $\mathcal{S}_t^+ = \emptyset$ **then**
 5:       Declare infeasibility
 6:    **end if**
 7:    $\mathbf{x}_t, \mathbf{z}_t \leftarrow \arg\max_{(\mathbf{x}, \mathbf{z}) \in \mathcal{S}_t^+ \cap \mathcal{P}_t^+} u_{F,t}(\mathbf{x}, \mathbf{z})$   // Eq. 4.6
 8:    $h_t \leftarrow \arg\max_{h \in \mathcal{F}} \bar{r}_{h,t}(\mathbf{x}_t, \mathbf{z}_t)$        // Def. 4.7
 9:    **if** $h_t = f$ **then**
10:       $\bar{\mathbf{z}}_t \leftarrow \arg\max_{\mathbf{z} \in \mathcal{S}_{\text{lo},t}^+(\mathbf{x}_t)} u_{f,t}(\mathbf{x}_t, \mathbf{z})$     // Eq. 4.5
11:       **if** $\sigma_{f,t-1}(\mathbf{x}_t, \bar{\mathbf{z}}_t) > \sigma_{f,t-1}(\mathbf{x}_t, \mathbf{z}_t)$ **then**
12:          $\mathbf{z}_t \leftarrow \bar{\mathbf{z}}_t$        // Eq. 4.8
13:       **end if**
14:    **end if**
15:    $\mathcal{D}_{h_t,t} \leftarrow \mathcal{D}_{h_t,t-1} \cup \{y_{h_t}(\mathbf{x}_t, \mathbf{z}_t)\}$
16:    Update GP posterior belief: $\mu_{h_t,t}, \sigma_{h_t,t}$
17:    Update trusted sets $\mathcal{S}_t^+, \mathcal{P}_t^+$      // Defs. 4.3 and 4.5
18: **end for**

---

Note the trusted feasible set extends the constraint regret bounds of constrained BO methods in Xu et al. (2023) and Nguyen et al. (2023) to all points in the trusted set compared to only on the query point, so as to facilitate combination with the trusted set of optimal lower-level solutions for bilevel optimization with constraints.

### 4.1. Trusted set of optimal lower-level solutions

We define another trusted set to approximate the unknown set of optimal lower-level solutions, using confidence bounds and estimated optimal lower-level solutions. Points in this trusted set have lower-level objective regret bounds, allowing us to quantify possible lower-level suboptimality.

**Definition 4.5** (Trusted set of optimal lower-level solutions). Let the trusted set of optimal lower-level solutions be

$$\mathcal{P}_t^+ \triangleq \{(\mathbf{x}, \mathbf{z}) \in \mathcal{S}_{\text{lo},t}^+ \mid u_{f,t}(\mathbf{x}, \mathbf{z}) \geq l_{f,t}(\mathbf{x}, \bar{\mathbf{z}}_t(\mathbf{x}))\}, \quad (4.4)$$

where $\mathcal{S}_{\text{lo},t}^+ \triangleq \{(\mathbf{x}, \mathbf{z}) \in \mathcal{X} \times \mathcal{Z} \mid u_{c,t}(\mathbf{x}, \mathbf{z}) \geq 0 \ \forall c \in \mathcal{C}_{\text{lo}}\}$ is the trusted set of feasible solutions w.r.t. lower-level constraints, and

$$\bar{\mathbf{z}}_t(\mathbf{x}) \triangleq \arg\max_{\mathbf{z} \in \mathcal{S}_{\text{lo},t}^+(\mathbf{x})} u_{f,t}(\mathbf{x}, \mathbf{z}), \quad (4.5)$$

is the estimated optimal lower-level solution at $\mathbf{x}$.

**Lemma 4.6.** *Let $\delta \in (0, 1)$, with probability at least $1 - \delta$, $\forall(\mathbf{x}, \mathbf{z}) \in \mathcal{P}_t^+$, the lower-level objective regret is upper bounded,*

$$r_{f,t}(\mathbf{x}, \mathbf{z}) \leq \mathbb{1}_{\mathbf{z} \neq \bar{\mathbf{z}}_t(\mathbf{x})} 2\beta_t^{1/2}\sigma_{f,t-1}(\mathbf{x}, \bar{\mathbf{z}}_t(\mathbf{x}))$$
$$+ 2\beta_t^{1/2}\sigma_{f,t-1}(\mathbf{x}, \mathbf{z}).$$

The proof is given in Appendix C.2. Sampling from $\mathcal{P}_t^+$ guarantees an upper-bounded lower-level objective regret, and points outside of the trusted set $\mathcal{P}_t^+$ are highly unlikely to be lower-level optimal.

The trusted set $\mathcal{P}_t^+$ allows multiple lower-level solutions to correspond to an upper-level variable, effectively managing multiple lower-level solutions and noisy observations. Moreover, we can handle infeasible lower-level problems as the trusted set $\mathcal{P}_t^+$ filters out highly probable infeasible points via the set $\mathcal{S}_{\text{lo},t}^+$, which can eliminate upper-level points that are infeasible at the lower-level, ensuring only probable feasible solutions are considered during optimization.

$\epsilon$-**optimal lower-level solutions**. In some scenarios, it may be desirable to consider $\epsilon$-optimal lower-level solutions feasible, as it is common for real-world agents to operate suboptimally. This allows us to account for practical limitations where perfect lower-level optimization may not be achievable, for example, due to noise or the cost of querying. In this case, we can relax the condition in Definition 4.5 to allow $\epsilon$-optimal lower-level solutions to remain in the trusted set by defining $\mathcal{P}_t^\epsilon \triangleq \{(\mathbf{x}, \mathbf{z}) \mid u_{f,t}(\mathbf{x}, \mathbf{z}) + \epsilon \geq l_{f,t}(\mathbf{x}, \bar{\mathbf{z}}_t(\mathbf{x}))\}$, and extending the regret bound in Lemma 4.6 to $r_{f,t}(\mathbf{x}, \mathbf{z}) \leq \epsilon + \mathbb{1}_{\mathbf{z} \neq \bar{\mathbf{z}}_t(\mathbf{x})} 2\beta_t^{1/2}\sigma_{f,t-1}(\mathbf{x}, \bar{\mathbf{z}}_t(\mathbf{x})) + 2\beta_t^{1/2}\sigma_{f,t-1}(\mathbf{x}, \mathbf{z})$.

## 4.2. Query point selection

We reduce the search space to $\mathcal{S}_t^+ \cap \mathcal{P}_t^+$. Points in this search space have upper-bounded instantaneous regrets on constraints and lower-level objective with high probability, according to Lemmas 4.4 and 4.6. The query point at time $t$ is sampled from the reduced search space and chosen w.r.t. the upper confidence bound of the upper-level objective $u_{F,t}$, as shown in Figure 2a and defined as,

$$\mathbf{x}_t, \mathbf{z}_t \triangleq \arg\max_{(\mathbf{x}, \mathbf{z}) \in \mathcal{S}_t^+ \cap \mathcal{P}_t^+} u_{F,t}(\mathbf{x}, \mathbf{z}). \tag{4.6}$$

## 4.3. Function query

In the decoupled case, a function query $h_t$ is selected at each timestep $t$ for evaluation. We follow the function query selection in Definition 4.7, and Lemma 4.8 provides an instantaneous regret bound on the query $(\mathbf{x}_t, \mathbf{z}_t)$.

**Definition 4.7** (Function query)**.** Let the function query $h_t$ selected at each timestep $t$ be

$$h_t \triangleq \arg\max_{h \in \mathcal{F}} \bar{r}_{h,t}(\mathbf{x}_t, \mathbf{z}_t), \tag{4.7}$$

where $\mathcal{F} \triangleq \{F, f\} \cup \mathcal{C}_{\text{up}} \cup \mathcal{C}_{\text{lo}}$. The estimated regrets are defined using confidence intervals and $\bar{\mathbf{z}}_t$ from Equation 4.5,

$$\bar{r}_{h',t}(\mathbf{x}_t, \mathbf{z}_t) \triangleq 2\beta_t^{1/2}\sigma_{h',t-1}(\mathbf{x}_t, \mathbf{z}_t), \ \forall h' \in \mathcal{F}/\{f\},$$

$$\bar{r}_{f,t}(\mathbf{x}_t, \mathbf{z}_t) \triangleq \mathbb{1}_{\mathbf{z} \neq \bar{\mathbf{z}}_t(\mathbf{x}_t)} 2\beta_t^{1/2}\sigma_{f,t-1}(\mathbf{x}_t, \bar{\mathbf{z}}_t(\mathbf{x}_t))$$
$$+ 2\beta_t^{1/2}\sigma_{f,t-1}(\mathbf{x}_t, \mathbf{z}_t).$$

The estimated lower-level objective regret $\bar{r}_{f,t}(\mathbf{x}_t, \mathbf{z}_t)$ has an additional term, $\sigma_{f,t-1}(\mathbf{x}_t, \bar{\mathbf{z}}_t(\mathbf{x}_t))$, compared to other regret components. This additional term increases $\bar{r}_{f,t}$ when the estimated lower-level solution is highly uncertain, corresponding to the intuitive need for more frequent queries of the lower-level objective, especially since most existing bilevel BO methods rely on global optimization of the lower level.

**Conditional reassignment of $\mathbf{z}_t$ for lower-level objective query.** When the lower-level objective function $f$ is selected for query, we want to reduce lower-level objective regret effectively. Thus, as in Figure 2b, the lower-level variable to query, $\mathbf{z}_t$, has to be reassigned on the following condition,

$$\text{If } h_t = f \text{ and } \sigma_{f,t-1}(\mathbf{x}_t, \bar{\mathbf{z}}_t(\mathbf{x}_t)) \geq \sigma_{f,t-1}(\mathbf{x}_t, \mathbf{z}_t),$$
$$\mathbf{z}_t \leftarrow \bar{\mathbf{z}}_t(\mathbf{x}_t). \tag{4.8}$$

Without reassignment, $f$ will only be queried at $(\mathbf{x}_t, \mathbf{z}_t)$ and the term $\sigma_{f,t-1}(\mathbf{x}_t, \bar{\mathbf{z}}_t(\mathbf{x}_t))$ would remain large even after repeated queries to $f$. This reassignment encourages exploration of the lower-level objective, replacing the repeated lower-level optimization required by existing methods.

**Lemma 4.8.** *Let $\delta \in (0, 1)$, with probability at least $1 - \delta$, following the function query selection in Definition 4.7 and reassignment of query point in Equation 4.8, the instantaneous regret for the query point $(\mathbf{x}_t, \mathbf{z}_t)$ at time $t \geq 1$ is upper bounded by,*

$$r_t \leq 4\beta_t^{1/2} \max_{h \in \mathcal{F}} \sigma_{h,t-1}(\mathbf{x}_t, \mathbf{z}_t).$$

The proof is in Appendix C.3. By Lemma C.4, we also see $\max_{h \in \mathcal{F}} \bar{r}_{h,t}(\mathbf{x}_t, \mathbf{z}_t) \geq r_t$. Thus, $\max_{h \in \mathcal{F}} \bar{r}_{h,t}(\mathbf{x}_t, \mathbf{z}_t)$ is the upper regret bound at query point $(\mathbf{x}_t, \mathbf{z}_t)$, where a large estimated regret $\bar{r}_{h,t}(\mathbf{x}_t, \mathbf{z}_t)$ suggests that function $h$ affects $r_t$ significantly. Since $\bar{r}_{h,t}$ comprises of $\sigma_{h,t-1}$, selecting the $\arg\max_{h \in \mathcal{F}} \bar{r}_{h,t}$ in Definition 4.7 can also be seen as selecting the most uncertain function at $(\mathbf{x}_t, \mathbf{z}_t)$ to query.

## 4.4. Regret bound

The cumulative regret bound of Algorithm 1 is shown in Theorem 4.9 and proven in Appendix C.4 using Lemma 4.8.

**Theorem 4.9.** *Let $\delta \in (0, 1)$ and $\beta_t \triangleq 2\log(|\mathcal{F}||\mathcal{X}||\mathcal{Z}|t^2\pi^2/6\delta)$. With probability of at least $1 - \delta$, Algorithm 1 has a cumulative regret bound of*

$$R_T \leq \sqrt{4T|\mathcal{F}|\beta_T \max_{h \in \mathcal{F}} C_h \gamma_{h,T}},$$

*where $C_h \triangleq 8/\log(1 + \sigma_h^{-2})$, and $\gamma_{h,T}$ is the maximum information gain from noisy observations of $h$ at $(\mathbf{x}_t, \mathbf{z}_t)$, $\forall t \in [T]$.*

The regret bound is related to the maximum information gain across all functions in $\mathcal{F}$. Our regret bound has a larger

constant than the regret bound for constrained Bayesian optimization in Nguyen et al. (2023), as the lower-level objective regret has a larger upper bound than constraint regret. This highlights the increased difficulties of optimizing bilevel problems, where suboptimal lower-level solutions can hinder upper-level optimization, making bilevel optimization more challenging than standard constrained optimization.

The cumulative regret bound of BILBO is sublinear as $\gamma_{h,T}$ is sublinear for common kernels including Squared Exponential and Matérn kernels (Srinivas et al., 2010). The sublinear cumulative regret guarantees convergence to the optimal solution as $R_T/T \to 0$ as $T \to \infty$.

Lemma 4.10 gives a simple regret bound for the estimator,

$$\hat{\mathbf{x}}_T, \hat{\mathbf{z}}_T \triangleq \arg \min_{(\mathbf{x}_t, \mathbf{z}_t) \in \{(\mathbf{x}_{t'}, \mathbf{z}_{t'})\}_{t' \in [T]}} \max_{h \in \mathcal{F}} \bar{r}_{h,t}(\mathbf{x}_t, \mathbf{z}_t). \tag{4.9}$$

**Lemma 4.10.** *Let $\delta \in (0, 1)$, with probability at least $1 - \delta$, $T \geq 1$, and $\beta_T \triangleq 2 \log(|\mathcal{F}||\mathcal{X}||\mathcal{Z}|T^2\pi^2/6\delta)$, the estimator $(\hat{\mathbf{x}}_T, \hat{\mathbf{z}}_T)$, defined in Equation 4.9, has a simple regret bound of*

$$r_T \leq \sqrt{4|\mathcal{F}|\beta_T \max_{h \in \mathcal{F}} C_h \gamma_{h,T}/T}.$$

This follows as the simple regret of $(\hat{\mathbf{x}}_T, \hat{\mathbf{z}}_T)$ is upper bounded by the average regret bound in Lemma 4.8 across timesteps. The detailed proof is in Appendix C.5.

# 5. Experiments

We evaluate the performance of BILBO on 4 synthetic and 2 real-world problems. We introduce 2 baselines for comparison: "TrustedRand" and "Nested". TrustedRand randomly samples query points from trusted sets to assess the trusted set's contribution to the overall performance of BILBO. Nested optimizes the upper- and lower-level problems separately and serves as a baseline for nested BO approaches like in Kieffer et al. (2017), approximating lower-level gradients. More details on TrustedRand and Nested are in Appendix D.1. Note Nested cannot handle constraints and is not compared in experiments with constraints (SMD12 and Chemical). The very recent parallel work by Ekmekcioglu et al. (2024) is not compared as code is not provided.

Algorithms are implemented using GpyTorch (Gardner et al., 2018). All experiments, except Nested, are initialized with 3 observations on each function. Nested requires more initial observations of lower-level functions as the upper-level objective is only evaluated at the estimated lower-level solution. We allow for this to enable comparisons, which also highlights the sample inefficiency of nested methods. All observations are noisy with $\sigma_n = 0.01$, and outputs are normalized to have mean 0 and standard deviation 1. We

discretize the search space using a uniformly-spaced grid to facilitate representation of trusted sets. BILBO queries only one function per iteration, while TrustedRand queries all function at each iteration. For comparison, the estimator is chosen as $\arg \max_{(\mathbf{x},\mathbf{z}) \in \mathcal{S}_t^+ \cap \mathcal{P}_t^+} \mu_{F,t}(\mathbf{x}, \mathbf{z})$ for BILBO and TrustedRand, and $\arg \max_{\mathbf{x} \in \mathcal{X}} \mu_{F,t}(\mathbf{x})$ for Nested. Additional implementation details are in Appendix D.2.

Results are averaged over 5 runs and performance is compared by examining the instantaneous regret against query count with 95% confidence intervals. The instantaneous regret in this section is calculated as the sum of each function's instantaneous regret ($\sum_{h \in \mathcal{F}} r_{h,t}$) to provide intuitive comparison across different methods. Initial observations are included in the number of queries, indicated by a small gap in the regret plots before estimations begin.

## 5.1. Synthetic problems

The synthetic problems were selected to cover a variety of scenarios, including conflicting interactions, convex or multimodal functions, and active constraints.

**BraninHoo+GoldsteinPrice** has the Branin-Hoo function as upper-level objective $F$ and the Goldstein-Price function as the lower-level objective $f$ (Picheny et al., 2013). Both functions are non-convex and multimodal. The dimensions $d_\mathcal{X}$ and $d_\mathcal{Z}$ are both 1, which facilitates visualization of the models and queries. Both dimensions were discretized into 100 points. The Branin-Hoo function has 3 optimal points, but there is only 1 optimal bilevel solution when constrained by lower-level Goldstein-Price optimal solutions.

BILBO outperforms the other two methods by a substantial margin, as seen in Figure 4a, where it converges to the optimal bilevel solution within 150 queries. Nested and TrustedRand converge equally slowly. For Nested, the predicted lower-level solutions may be suboptimal because the lower-level solver cannot handle multimodal functions and noisy observations effectively. For TrustedRand, random queries might have led to uninformative points being sampled. The challenging multimodal characteristic of the functions also means that sampling in informative areas is integral for this problem, which BILBO successfully manages to do.

Figures 3a and 3b show the upper- and lower-level objective function respectively, with optimal lower-level solutions (yellow dots), and Figures 3c to 3f show BILBO's inner workings. BILBO converged to the optimal solution, and the surrogate models effectively captured the overall landscape of both functions in Figures 3c and 3d, where predicted lower-level solutions (yellow crosses) are close to optimal lower-level solutions, especially in regions where upper-level objective value is high. Queries selected over iterations from the upper- and lower-level objective functions are in Figures 3e and 3f respectively, with darker colors indicating

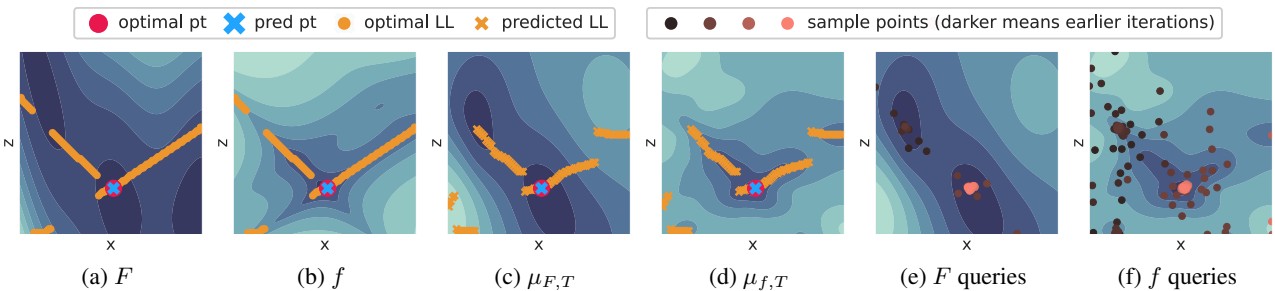

(a) $F$     (b) $f$     (c) $\mu_{F,T}$     (d) $\mu_{f,T}$     (e) $F$ queries     (f) $f$ queries

Figure 3: BraninHoo+GoldsteinPrice experiment details. LL refers to lower-level. (a) Upper-level objective, Branin-Hoo. (b) Lower-level objective, Goldstein-Price. (c) BILBO's upper-level estimate. (d) BILBO's lower-level estimate. (e-f) BILBO's iterative queries.

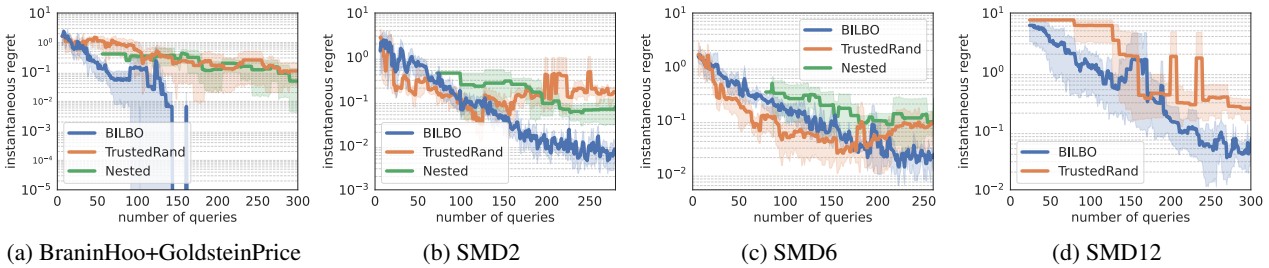

(a) BraninHoo+GoldsteinPrice     (b) SMD2     (c) SMD6     (d) SMD12

Figure 4: Instantaneous regrets (log-scale) over number of queries, averaged over 5 runs, for synthetic experiments.

samples from earlier iterations, and they mostly clustered around two probable optimal solutions. BILBO sampled the objective functions in the top-left region until it ascertained the absence of optimal solutions and converged on the actual optimal solution, demonstrating its effectiveness.

**SMD2**, **SMD6**, and **SMD12** are adapted from the SMD suite of test problems for bilevel optimization (Sinha et al., 2014). Details of implementation are in Appendix D.3. The input dimension of the test problems is set to 5, with $d_{\mathcal{X}}$ being 2 and $d_{\mathcal{Z}}$ being 3. The difficulty increases in the order of SMD2, SMD6, SMD12. SMD2 has convex functions and conflicting interactions, where improving the lower-level estimate worsens the upper-level objective value. This requires the algorithm to predict lower-level optimal solutions accurately to obtain the optimal bilevel solution. SMD6 also has convex functions and conflicting interactions, but with multiple lower-level optimal solutions at each upper-level point (i.e., a convex valley). An algorithm must concurrently estimate multiple lower-level optimal solutions and identify the point that optimizes the upper-level objective. Finally, SMD12 is the most challenging problem from the SMD suite, with both levels having 3 active constraints, where the optimal solution is on the boundary of the constraints. There are also multiple optimal solutions at the lower level.

Results of the SMD experiments are shown in Figures 4b to 4d. For SMD2, BILBO outperforms both TrustedRand and Nested. While TrustedRand's regret decreased quickly

at the start, its rate of decrease diminishes over time, likely because random queries are initially informative but become less effective as the process continues. For SMD6, BILBO has the smallest regret after around 250 steps. Nested is unable to handle multiple lower-level optimal solutions, as it predicts only one lower-level solution for each upper-level point. In comparison, the trusted sets allow multiple optimal lower-level estimates for both BILBO and TrustedRand. For SMD12, BILBO converges faster than TrustedRand. With 8 functions in the SMD12 problem, the decoupled setting becomes more crucial for sample efficiency. The faster convergence of BILBO demonstrates the effectiveness of our function query strategy in selecting more informative functions to query. The presence of active constraints did not appear to pose any difficulties for BILBO as well.

### 5.2. Real-world problems

**Energy.** We simulated a bilevel energy market problem, where energy providers bid to supply an amount of electricity at the upper level to maximize profits over three time periods. At the lower level, they optimize their operations, considering costs, demand responses to prices, and their ability to meet changing demands. There are 2 upper-level variables: price and quantity of electricity to bid, and 2 lower-level variables: the ramp limit for one power plant and the maximum power output at each period for another power plant. Lower-level variables affect overall optimal

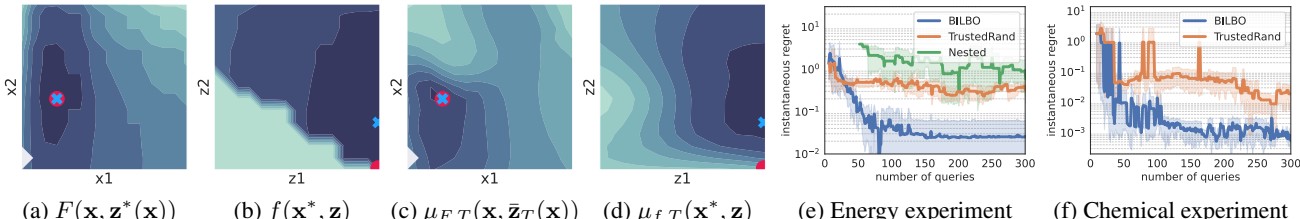

(a) $F(\mathbf{x}, \mathbf{z}^*(\mathbf{x}))$    (b) $f(\mathbf{x}^*, \mathbf{z})$    (c) $\mu_{F,T}(\mathbf{x}, \bar{\mathbf{z}}_T(\mathbf{x}))$    (d) $\mu_{f,T}(\mathbf{x}^*, \mathbf{z})$    (e) Energy experiment    (f) Chemical experiment

Figure 5: Real-world experiments. (a-b) Functions from energy experiment and (c-d) BILBO outputs, with optimal solution (red dot) and predicted solution (blue cross). (e-f) Regret plots in log-scale for energy and chemical experiments respectively.

dispatch of electricity and the dispatching of three power plants was simulated using PyPSA (Brown et al., 2018). We formulated the lower-level objective as a function of simulation outputs, to seek the lowest cost of electricity generation while incorporating penalties to reduce wear and tear or other auxiliary concerns. More details in Appendix D.4.

Figure 5e shows BILBO outperforms the other methods, with its regret decreasing the fastest. Refer to Figures 5c and 5d for surrogate models learned after one run of BILBO. Figures 5a and 5c, respectively, show the upper-level objective $F$ at optimal lower-level solutions and the estimated upper-level objective $\mu_{F,T}$ at estimated lower-level solutions. $\mu_{F,T}$ approximates $F$ well, especially at regions with high $F$ values, and correctly predicted the optimal bilevel solution. Figures 5b and 5d, respectively, show the lower-level objective $f$ and the estimated lower-level objective $\mu_{f,T}$ at the optimal upper-level variable. At this upper-level variable, $\mu_{f,T}$ captures the general trend of $f$, where optimal points are on the right. However, the optimal lower-level solution is at a boundary with high discontinuity. The surrogate model was unable to model this large step and predicted a suboptimal lower-level solution, resulting in an empirical asymptotic regret bound in Figure 5e, compounded by noisy observations. This may be mitigated by adding a constraint function to represent the discontinuity in the lower-level objective, as BILBO has shown the capability to handle active constraints effectively in previous synthetic experiments.

**Chemical.** Chemical processes in industries such as pharmaceuticals, petrochemicals, and food production often involve multiple stages, each requiring parameter optimization. Bilevel optimization simplifies this by dividing the overall process into smaller, more manageable problems, while still accounting for the interactions between different stages. We used COCO simulator to simulate carbonylation of Di-Methyl Ether (DME) to Methyl Acetate, adapted from the flowsheet provided by ChemSep. The upper-level problem focuses on maximizing the yield of Methyl Acetate at 99.9% purity through a distillation column, which takes in a reaction mixture comprising Methyl Acetate, unreacted DME, and by-products. These are outputs from the lower-level optimization, which involves carbonylation of DME

to produce Methyl Acetate in a reactor. Additionally, an upper-level constraint is included to ensure a suitable temperature range for chemicals to be in their correct states. There is 1 upper-level variable: the number of levels in the distillation column, and 3 lower-level variables: temperature of the reactor, number of heating tubes, and the diameter of heating tubes. More details are in Appendix D.5. Results are in Figure 5f where BILBO outperforms TrustedRand, highlighting the potential efficiency and effectiveness of BILBO in optimizing complex industrial operations.

Additional discussions on the computational complexity and time efficiency of BILBO can be found in Appendix E.

## 6. Future Work

We have shown theoretically and empirically that BILBO is a regret-bounded, sample efficient algorithm for noisy, constrained, and derivative-free bilevel optimization. A key direction for future work is improving scalability to high-dimensional spaces, which is a common challenge in BO.

We currently model upper-level objective $F$ over $\mathcal{X} \times \mathcal{Z}$, but this can be memory inefficient as many lower-level variables are suboptimal and irrelevant. A more efficient approach could involve directly modeling $F(\mathbf{x}, \mathbf{z}^*(\mathbf{x}))$, reducing the dimension of the surrogate model from $d_{\mathcal{X}} \times d_{\mathcal{Z}}$ to $d_{\mathcal{X}}$. This poses another challenge: incorporating the uncertainty associated with the optimality of the lower-level solution into the uncertainty of the upper-level objective value.

Adaptive discretization (Shekhar & Javidi, 2018) may also reduce computational complexity by reducing the effective dimension of the explored space. Discretization strategies could be integrated with trusted sets, for example concentrating the discretizations within the trusted sets. Approximate surrogate models (Calandriello et al., 2019) is another possible direction for scalability while preserving confidence bound estimates. The theoretical work presented in this paper could be extended to approximate surrogate models.

The representation of trusted sets will need to scale effectively to higher dimensions as well. Possible approaches could be via sampling strategies like Latin Hypercube Sam-

pling (McKay et al., 2000) for efficient point representation in high-dimensional spaces or using hyperrectangles to represent the trusted set efficiently (Eriksson et al., 2019).

Future work can extend our discretized implementation to continuous domains, which should be feasible with a scalable trusted set representation. Our theoretical results can also generalize to continuous settings with minor modifications to $\beta_t$ and additional assumptions, following Chowdhury & Gopalan (2017).

## 7. Conclusion

We introduced BILBO, a novel bilevel BO algorithm that optimizes the upper- and lower-levels simultaneously. BILBO samples from confidence-bounds based trusted sets to bound lower-level suboptimality, and encourages lower-level exploration via conditional reassignment of the query, replacing the repeated lower-level optimizations required by existing methods. We show theoretically that BILBO has a sublinear regret bound, and our experiments demonstrate empirically that BILBO outperforms other bilevel optimization baselines, especially in problems with many non-convex functions. BILBO is a significant step towards a general bilevel solver, which will enable applications to complex real-world bilevel problems involving blackbox functions.

## Acknowledgments

This research is supported by the National Research Foundation (NRF), Prime Minister's Office, Singapore, under its Campus for Research Excellence and Technological Enterprise (CREATE) programme. The Mens, Manus, and Machina (M3S) is an interdisciplinary research group (IRG) of the Singapore MIT Alliance for Research and Technology (SMART) centre.

## Impact Statement

This paper presents work whose goal is to advance the field of Machine Learning. There are many potential societal consequences of our work, none which we feel must be specifically highlighted here.

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

# A. Table of Notations

## Bilevel definitions

**Upper-level**

| | | | |
|---|---|---|---|
| $\mathbf{x}$ | Upper-level variable |
| $\mathcal{X}$ | Domain of $\mathbf{x}$ |
| $d_{\mathcal{X}}$ | Dimension of $\mathbf{x}$ |
| $F$ | Upper-level objective function |
| $\mathcal{C}_{\text{up}}$ | Set of upper-level constraint functions |
| $\mathbf{x}_t$ | Selected upper-level variable to query at time $t$ |
| $\hat{\mathbf{x}}_T$ | Estimated optimal upper-level variable at time $T$ |

**Lower-level**

| | |
|---|---|
| $\mathbf{z}$ | Lower-level variable |
| $\mathcal{Z}$ | Domain of $\mathbf{z}$ |
| $d_{\mathcal{Z}}$ | Dimension of $\mathbf{z}$ |
| $f$ | Lower-level objective function |
| $\mathcal{C}_{\text{lo}}$ | Set of lower-level constraint functions |
| $\mathbf{z}_t$ | Selected lower-level variable to query at time $t$ |
| $\hat{\mathbf{z}}_t$ | Estimated optimal lower-level variable at time $T$ |

| | |
|---|---|
| $\mathcal{F}$ | Set of functions in a bilevel problem $\{F, f\} \cup \mathcal{C}_{\text{up}} \cup \mathcal{C}_{\text{lo}}$ |
| $h$ | Arbitrary function in $\mathcal{F}$ |
| $\mu_{h,t}(\mathbf{x}, \mathbf{z})$ | GP posterior mean at $(\mathbf{x}, \mathbf{z})$ for function $h$ at time $t$ |
| $\sigma_{h,t}(\mathbf{x}, \mathbf{z})$ | GP posterior standard deviation at $(\mathbf{x}, \mathbf{z})$ for function $h$ at time $t$ |
| $r_h(\mathbf{x}, \mathbf{z})$ | Instantaneous regret of function $h$ at $(\mathbf{x}, \mathbf{z})$ |
| $r_t$ | Instantaneous bilevel regret at time $t$ on query point $(\mathbf{x}_t, \mathbf{z}_t)$ |
| $R_T$ | Cumulative regret at time $T$ |
| $r_T$ | Simple bilevel regret at time $T$ based on $(\hat{\mathbf{x}}_t, \hat{\mathbf{z}}_t)$ |

## BILBO notations

| | |
|---|---|
| $u_{h,t}(\mathbf{x}, \mathbf{z})$ | Upper confidence bound of function $h$ at $(\mathbf{x}, \mathbf{z})$ (Defn. 4.1) |
| $l_{h,t}(\mathbf{x}, \mathbf{z})$ | Lower confidence bound of function $h$ at $(\mathbf{x}, \mathbf{z})$ (Defn. 4.1) |
| $\mathcal{S}_t^+$ | Trusted set of feasible solutions (Defn. 4.3) |
| $\mathcal{S}_{\text{lo},t}^+$ | Trusted set of feasible solutions w.r.t. only lower-level constraints (Defn. 4.5) |
| $\mathcal{P}_t^+$ | Trusted set of optimal lower-level solutions (Defn. 4.5) |
| $\bar{\mathbf{z}}_t(\mathbf{x})$ | Estimated optimal lower-level solution at $\mathbf{x}$ at timestep $t$ (Defn. 4.5) |
| $h_t$ | Selected function query (Defn. 4.7) |
| $\bar{r}_{h,t}$ | Estimated regret for function $h$ (Defn. 4.7) |

# B. More preliminaries details

## B.1. Closed-form posteriors of Gaussian Processes

For a GP defined as $\mathcal{GP}_h(m_h(\mathbf{xz}), k_h(\mathbf{xz}, \mathbf{xz}'))$ for a function $h$. The closed-form posterior mean is $\mu_{h,t-1}(\mathbf{xz}) \triangleq m_h(\mathbf{xz}) + \mathbf{k}_{h,t-1}(\mathbf{xz})^\top (\mathbf{K}_{h,t-1} + \sigma^2 \mathbf{I})^{-1}(\mathbf{y}_{h,t-1} - \mathbf{m}_{h,t-1})$ and variance $\sigma^2_{h,t-1}(\mathbf{xz}) \triangleq k_h(\mathbf{xz}, \mathbf{xz}) - \mathbf{k}_{h,t-1}^\top (\mathbf{K}_{h,t-1} + \sigma^2 \mathbf{I})^{-1}\mathbf{k}_{h,t-1}^{-1}$ where $\mathbf{m}_{h,t-1} \triangleq [m_h(\mathbf{x}, \mathbf{z})]_{\mathbf{xz} \in \mathbf{xz}_{:t-1}}$, $\mathbf{k}_{h,t-1}(\mathbf{xz}) \triangleq [k_h(\mathbf{xz}, \mathbf{xz}')]_{\mathbf{xz}' \in \mathbf{xz}_{:t-1}}$, and $\mathbf{K}_{h,t-1} \triangleq [k_h(\mathbf{xz}, \mathbf{xz}')]_{\mathbf{xz}, \mathbf{xz}' \in \mathbf{xz}_{:t-1}}$.

## B.2. Maximum information gain

Maximum information gain, $\gamma_{h,t}$, on a function $h$, where $d \triangleq d_{\mathcal{X}} + d_{\mathcal{Z}}$ and $T(h)$ contains the timesteps when function $h$ was selected for query, from Vakili et al. (2021):

- Squared Exponential kernel: $\mathcal{O}(\log^{d+1}(T(h)))$

- Matérn kernels with $\nu > \frac{1}{2}$: $\mathcal{O}(T^{\frac{d}{2\nu+d}} \log^{\frac{2\nu}{2\nu+d}}(T(h)))$

# C. Proofs

## C.1. Proof of Lemma 4.4

*Proof.* $\forall c \in \mathcal{C}_{\text{up}} \cup \mathcal{C}_{\text{lo}}, (\mathbf{x}, \mathbf{z}) \in \mathcal{S}_t^+$,

$$
\begin{aligned}
r_{c,t}(\mathbf{x}, \mathbf{z}) &\triangleq \max(0, -c(\mathbf{x}, \mathbf{z})) && \text{from Equation 3.7} \\
&\leq \max(0, -l_{c,t}(\mathbf{x}, \mathbf{z})) && \text{from Corollary 4.2} \\
&\leq \max(0, u_{c,t}(\mathbf{x}, \mathbf{z}) - l_{c,t}(\mathbf{x}, \mathbf{z})) && \text{from } (\mathbf{x}, \mathbf{z}) \in \mathcal{S}_t^+ \\
&\leq 2\beta_t^{1/2} \sigma_{c,t-1}(\mathbf{x}, \mathbf{z}). && \text{from Definition 4.1}
\end{aligned}
$$

$\square$

## C.2. Proof of Lemma 4.6

**Lemma C.1.** $\forall \mathbf{x} \in \{\mathbf{x} \mid (\mathbf{x}, \mathbf{z}) \in \mathcal{P}_t^+\}$,

$$u_{f,t}(\mathbf{x}, \bar{\mathbf{z}}_t(\mathbf{x})) \geq u_{f,t}(\mathbf{x}, \mathbf{z}^*(\mathbf{x})), \tag{C.1}$$

*where $\bar{\mathbf{z}}_t(\mathbf{x}) \triangleq \arg\max_{\mathbf{z} \in \mathcal{S}_{lo,t}^+(\mathbf{x})} u_{f,t}(\mathbf{x}, \mathbf{z})$ is the estimated optimal lower-level solution at $\mathbf{x}$, and $\mathbf{z}^*(\mathbf{x})$ is the actual optimal lower-level solution at $\mathbf{x}$.*

*Proof.* By definition of $\bar{\mathbf{z}}_t(\mathbf{x})$, $\forall (\mathbf{x}, \mathbf{z}) \in \mathcal{S}_{lo,t}^+, u_{f,t}(\mathbf{x}, \bar{\mathbf{z}}_t(\mathbf{x})) \geq u_{f,t}(\mathbf{x}, \mathbf{z})$.

Let $\mathcal{S}_{\text{lo}} \triangleq \{(\mathbf{x}, \mathbf{z}) \mid c(\mathbf{x}, \mathbf{z}) \geq 0 \;\; \forall c \in \mathcal{C}_{\text{lo}}\}$ be the unknown set of feasible solutions w.r.t. lower-level constraints. Then, $(\mathbf{x}, \mathbf{z}^*(\mathbf{x})) \in \mathcal{S}_{lo,t}^+$, because $(\mathbf{x}, \mathbf{z}^*(\mathbf{x})) \in \mathcal{S}_{\text{lo}}$ by definition and $\mathcal{S}_{\text{lo}} \subseteq \mathcal{S}_{lo,t}^+$ from Corollary 4.2.

Finally, by Definition 4.5 of $\mathcal{P}_t^+$, $\mathcal{P}_t^+ \subseteq \mathcal{S}_{lo,t}^+$.

$\square$

**Main proof** for instantaneous regret bound on $f$ in Lemma 4.6.

*Proof.* $\forall (\mathbf{x}, \mathbf{z}) \in \mathcal{P}_t^+$,

$$
\begin{aligned}
r_{f,t}(\mathbf{x}, \mathbf{z}) &= f(\mathbf{x}, \mathbf{z}^*(\mathbf{x})) - f(\mathbf{x}, \mathbf{z}) && \text{from Equation 3.6} \\
&\leq u_{f,t}(\mathbf{x}, \mathbf{z}^*(\mathbf{x})) - l_{f,t}(\mathbf{x}, \mathbf{z}) && \text{from Corollary 4.2} \\
&\leq u_{f,t}(\mathbf{x}, \bar{\mathbf{z}}_t(\mathbf{x})) - l_{f,t}(\mathbf{x}, \mathbf{z}). && \text{from Lemma C.1}
\end{aligned}
$$

For $\mathbf{z} = \bar{\mathbf{z}}_t(\mathbf{x})$,

$$
\begin{aligned}
r_{f,t}(\mathbf{x}, \mathbf{z}) &\leq u_{f,t}(\mathbf{x}, \bar{\mathbf{z}}_t(\mathbf{x})) - l_{f,t}(\mathbf{x}, \bar{\mathbf{z}}_t(\mathbf{x})) \\
&= 2\beta_t^{1/2}\sigma_{f,t-1}(\mathbf{x}, \bar{\mathbf{z}}_t(\mathbf{x})) && \text{from Definition 4.1}
\end{aligned}
$$

and for $\mathbf{z} \neq \bar{\mathbf{z}}_t(\mathbf{x})$,

$$
\begin{aligned}
r_{f,t}(\mathbf{x}, \mathbf{z}) &\leq u_{f,t}(\mathbf{x}, \bar{\mathbf{z}}_t(\mathbf{x})) - l_{f,t}(\mathbf{x}, \mathbf{z}) \\
&\leq u_{f,t}(\mathbf{x}, \bar{\mathbf{z}}_t(\mathbf{x})) - u_{f,t}(\mathbf{x}, \mathbf{z}) + 2\beta_t^{1/2}\sigma_{f,t-1}(\mathbf{x}, \mathbf{z}) && \text{from Definition 4.1} \\
&\leq u_{f,t}(\mathbf{x}, \bar{\mathbf{z}}_t(\mathbf{x})) - l_{f,t}(\mathbf{x}, \bar{\mathbf{z}}_t(\mathbf{x})) + 2\beta_t^{1/2}\sigma_{f,t-1}(\mathbf{x}, \mathbf{z}) && \text{from } (\mathbf{x}, \mathbf{z}) \in \mathcal{P}_t^+ \\
&= 2\beta_t^{1/2}\sigma_{f,t-1}(\mathbf{x}, \bar{\mathbf{z}}_t(\mathbf{x})) + 2\beta_t^{1/2}\sigma_{f,t-1}(\mathbf{x}, \mathbf{z}). && \text{from Definition 4.1}
\end{aligned}
$$

Combining both cases, we get the instantaneous regret for lower-level objective function as

$$
r_{f,t}(\mathbf{x}, \mathbf{z}) \leq \mathbb{1}_{\mathbf{z} \neq \bar{\mathbf{z}}_t(\mathbf{x})} 2\beta_t^{1/2}\sigma_{f,t-1}(\mathbf{x}, \bar{\mathbf{z}}_t(\mathbf{x})) + 2\beta_t^{1/2}\sigma_{f,t-1}(\mathbf{x}, \mathbf{z}).
$$

$\square$

### C.3. Proof of Lemma 4.8

**Lemma C.2.**
$$
(\mathbf{x}^*, \mathbf{z}^*) \in \mathcal{S}_t^+ \cap \mathcal{P}_t^+,
$$

*where $(\mathbf{x}^*, \mathbf{z}^*)$ is the optimal bilevel solution.*

*Proof.* Let the unknown feasible set be $\mathcal{S} \triangleq \{(\mathbf{x}, \mathbf{z}) \mid c(\mathbf{x}, \mathbf{z}) \geq 0 \;\; \forall c \in \mathcal{C}_{\text{up}} \cup \mathcal{C}_{\text{lo}}\}$. Since $(\mathbf{x}^*, \mathbf{z}^*) \in \mathcal{S}$ by definition and $\mathcal{S} \subseteq \mathcal{S}_t^+$ by Corollary 4.2, we have $(\mathbf{x}^*, \mathbf{z}^*) \in \mathcal{S}_t^+$.

Let unknown feasible set w.r.t. lower-level constraints be $\mathcal{S}_{\text{lo}} \triangleq \{(\mathbf{x}, \mathbf{z}) \mid c(\mathbf{x}, \mathbf{z}) \geq 0 \;\; \forall c \in \mathcal{C}_{\text{lo}}\}$. Similarly, we have $(\mathbf{x}^*, \mathbf{z}^*) \in \mathcal{S}_{\text{lo}} \subseteq \mathcal{S}_{\text{lo},t}^+$. Since $u_{f,t}(\mathbf{x}^*, \mathbf{z}^*) \geq f(\mathbf{x}^*, \mathbf{z}^*) \geq f(\mathbf{x}^*, \bar{\mathbf{z}}_t(\mathbf{x}^*)) \geq l_{f,t}(\mathbf{x}^*, \bar{\mathbf{z}}_t(\mathbf{x}^*))$, we have $(\mathbf{x}^*, \mathbf{z}^*) \in \mathcal{P}_t^+$.

$$
(\mathbf{x}^*, \mathbf{z}^*) \in \mathcal{S}_t^+ \text{ and } (\mathbf{x}^*, \mathbf{z}^*) \in \mathcal{P}_t^+ \Rightarrow (\mathbf{x}^*, \mathbf{z}^*) \in \mathcal{S}_t^+ \cap \mathcal{P}_t^+
$$

$\square$

**Lemma C.3.** *For some small $\delta > 0$, with probability at least $1 - \delta$, the instantaneous upper-level objective regret is upper bounded at the query point,*
$$
r_F(\mathbf{x}_t, \mathbf{z}_t) \leq 2\beta_t^{1/2}\sigma_{F,t-1}(\mathbf{x}_t, \mathbf{z}_t).
$$

*Proof.*

$$
\begin{aligned}
r_F(\mathbf{x}_t, \mathbf{z}_t) &\triangleq \max(0, F(\mathbf{x}^*, \mathbf{z}^*) - F(\mathbf{x}_t, \mathbf{z}_t)) && \text{from Equation 3.5} \\
&\leq \max(0, u_{F,t}(\mathbf{x}^*, \mathbf{z}^*) - l_{F,t}(\mathbf{x}_t, \mathbf{z}_t)) && \text{from Corollary 4.2} \\
&\leq \max_{(\mathbf{x}, \mathbf{z}) \in \mathcal{S}_t^+ \cap \mathcal{P}_t^+} u_{F,t}(\mathbf{x}, \mathbf{z}) - l_{F,t}(\mathbf{x}_t, \mathbf{z}_t) && \text{from Lemma C.2} \\
&= u_{F,t}(\mathbf{x}_t, \mathbf{z}_t) - l_{F,t}(\mathbf{x}_t, \mathbf{z}_t) && \text{from } \mathbf{x}_t, \mathbf{z}_t \triangleq \arg\max_{\mathcal{S}_t^+ \cap \mathcal{P}_t^+} u_{F,t} \\
&= 2\beta_t^{1/2}\sigma_{F,t-1}(\mathbf{x}_t, \mathbf{z}_t). && \text{from Definition 4.1}
\end{aligned}
$$

$\square$

**Lemma C.4.** *Given the estimated regret of the selected function query $h_t$ at the query point by Definition 4.7, the instantaneous regret $r_t$ is upper bounded,*
$$
r_t \leq \bar{r}_{h_t,t}(\mathbf{x}_t, \mathbf{z}_t).
$$

*Proof.* Given Definition 4.7, Lemma C.3, Lemma 4.4, and Lemma 4.6, $\forall h \in \mathcal{F}$, we can see that $\bar{r}_{h,t}(\mathbf{x}_t, \mathbf{z}_t) \geq r_h(\mathbf{x}_t, \mathbf{z}_t)$. Then,

$$\begin{aligned}
r_t &\triangleq \max_{h \in \mathcal{F}} r_h(\mathbf{x}_t, \mathbf{z}_t) && \text{from Equation 3.4}\\
&\leq \max_{h \in \mathcal{F}} \bar{r}_{h,t}(\mathbf{x}_t, \mathbf{z}_t)\\
&= \bar{r}_{h_t,t}(\mathbf{x}_t, \mathbf{z}_t).
\end{aligned}$$

$\square$

**Main proof** for instantaneous regret bound in Lemma 4.8

*Proof.* By Lemma C.4, if $h_t = f$,

$$\begin{aligned}
r_t &\leq \bar{r}_{f,t}(\mathbf{x}_t, \mathbf{z}_t)\\
&= \mathbb{1}_{\mathbf{z}_t \neq \bar{\mathbf{z}}_t(\mathbf{x}_t)} 2\beta_t^{1/2} \sigma_{f,t-1}(\mathbf{x}_t, \bar{\mathbf{z}}_t(\mathbf{x}_t)) + 2\beta_t^{1/2} \sigma_{f,t-1}(\mathbf{x}_t, \mathbf{z}_t) && \text{from Definition 4.7}\\
&\leq 4\beta_t^{1/2} \max(\sigma_{f,t-1}(\mathbf{x}_t, \bar{\mathbf{z}}_t(\mathbf{x}_t)), \sigma_{f,t-1}(\mathbf{x}_t, \mathbf{z}_t))\\
&= 4\beta_t^{1/2} \sigma_{f,t-1}(\mathbf{x}_t, \mathbf{z}_t),
\end{aligned}$$

where the last line holds because we reassign $\mathbf{z}_t \triangleq \bar{\mathbf{z}}_t(\mathbf{x}_t)$ if $\sigma_{f,t-1}(\mathbf{x}_t, \bar{\mathbf{z}}_t(\mathbf{x}_t)) \geq \sigma_{f,t-1}(\mathbf{x}_t, \mathbf{z}_t)$ as in Equation 4.8.

Else if $h_t \in \mathcal{F}/\{f\}$,

$$\begin{aligned}
r_t &\leq \bar{r}_{h_t,t}(\mathbf{x}_t, \mathbf{z}_t)\\
&= 2\beta_t^{1/2} \sigma_{h_t,t-1}(\mathbf{x}_t, \mathbf{z}_t)\\
&\leq 4\beta_t^{1/2} \sigma_{h_t,t-1}(\mathbf{x}_t, \mathbf{z}_t).
\end{aligned}$$

Combining, we obtain

$$\begin{aligned}
r_t &\leq 4\beta_t^{1/2} \sigma_{h_t,t-1}(\mathbf{x}_t, \mathbf{z}_t)\\
&\leq 4\beta_t^{1/2} \max_{h \in \mathcal{F}} \sigma_{h,t-1}(\mathbf{x}_t, \mathbf{z}_t).
\end{aligned}$$

$\square$

## C.4. Proof of Theorem 4.9

*Proof.* From Lemma 4.8 and by Cauchy-Schwarz inequality, we derive the cumulative regret as

$$\begin{aligned}
R_T^2 &\leq T \sum_{t=1}^{T} r_t^2\\
&\leq T \sum_{t=1}^{T} 16\beta_t \max_{h \in \mathcal{F}} \sigma_{h,t-1}^2(\mathbf{x}_t, \mathbf{z}_t)\\
&\leq 4T\beta_T \sum_{h \in \mathcal{F}} \sum_{t \in T(h)} 4\sigma_{h,t-1}^2(\mathbf{x}_t, \mathbf{z}_t)\\
&\leq 4T\beta_T \sum_{h \in \mathcal{F}} C_h \gamma_{h,T(h)}\\
&\leq 4T\beta_T \sum_{h \in \mathcal{F}} C_h \gamma_{h,T}\\
&\leq 4T|\mathcal{F}|\beta_T \max_{h \in \mathcal{F}} C_h \gamma_{h,T},
\end{aligned}$$

| experiment | length scale prior | $d_\mathcal{X}$ | $d_\mathcal{Z}$ | discrete points per dimension |
|---|---|---|---|---|
| BraninHoo+GoldsteinPrice | 0.2 | 1 | 1 | 100 |
| SMD2 | 0.7 | 2 | 3 | 25 |
| SMD6 | 0.2 | 2 | 3 | 25 |
| SMD12 | 0.4 | 2 | 3 | 16 |
| Energy | 0.4 | 2 | 2 | 15 |
| Chemical | 0.8 | 1 | 3 | 10 |

Table 1: Experiment parameters

where $T(h)$ contains the timesteps where function $h$ was queried, so $\gamma_{T(h)} \leq \gamma_T$, and

$$R_T \leq \sqrt{4T|\mathcal{F}|\beta_T \max_{h \in \mathcal{F}} C_h \gamma_{h,T}},$$

where $C_h \triangleq 8/\log(1 + \sigma_h^{-2})$, and $\gamma_{h,T}$ is the maximum information gain from noisy observations of $h$ at $(\mathbf{x}_t, \mathbf{z}_t), \forall t \in [T]$. The proof methodology follows Srinivas et al. (2010). □

### C.5. Proof of Lemma 4.10

*Proof.*

$$
\begin{aligned}
r_T &\leq \min_{(\mathbf{x}_t, \mathbf{z}_t) \in \{(\mathbf{x}_{t'}, \mathbf{z}_{t'})\}_{t' \in [T]}} \max_{h \in \mathcal{F}} \bar{r}_{h,t}(\mathbf{x}_t, \mathbf{z}_t) && \text{from Equation 4.9 and Lemma C.4} \\
&\leq \frac{1}{T} \sum_{t=1}^{T} \max_{h \in \mathcal{F}} \bar{r}_{h,t}(\mathbf{x}_t, \mathbf{z}_t) \\
&\leq \frac{1}{T} \sum_{t=1}^{T} 4\beta_t^{1/2} \max_{h \in \mathcal{F}} \sigma_{h,t-1}(\mathbf{x}_t, \mathbf{z}_t) && \text{from Appendix C.3} \\
&\leq \sqrt{4|\mathcal{F}|\beta_T \max_{h \in \mathcal{F}} C_h \gamma_{h,T}/T}. && \text{from Appendix C.4}
\end{aligned}
$$

□

## D. Experiment details

### D.1. Baseline details

**TrustedRand** implements a vanilla variant of the trusted sets $\mathcal{S}_t^+$ and $\mathcal{P}_t^+$, where mean $\mu$ is used instead of upper confidence bound $u$. Query points are then randomly sampled from the trusted set variants.

**Nested** uses the sequential least squares programming (SLSQP) optimizer for lower-level optimization, following Kieffer et al. (2017) and Dogan & Prestwich (2023), and BO with upper confidence bound acquisition function (Srinivas et al., 2010) at the upper level. The lower-level problem is solved to convergence at each upper-level query. Note that gradients are approximated for SLSQP, which can only work on continuous functions.

### D.2. Implementation details

GP with Matérn 5/2 kernel was used, and the GP hyperparameters were automatically tuned at each iteration using maximum likelihood estimation on the past observations. The hyperparameters include length scale and prior mean. The prior mean initialized to 0 for all experiments since the output is already normalized. The initial length scale and other parameters for each experiment are set according to Table 1. For SMD2, energy, and chemical experiment, we sampled from $\bar{\mathcal{P}}_t \triangleq \{(\mathbf{x}, \bar{\mathbf{z}}_t(\mathbf{x})) \, \forall \mathbf{x} \in \mathcal{X}\}$ instead of $\mathcal{P}_t^+$ as it was empirically found to be better.

## D.3. Edits to SMD2, SMD6, SMD12

The selected SMD problems were adapted so the input ranges from 0 to 1, and the outputs have a mean of 0 and standard deviation of 1, for parameters $p = 1, r = 1, q = 2$, while ensuring that their characteristics and optimal points remain the same. The upper- and lower-level objective functions of SMD each have 3 components. The following only records edits to the original SMD problems. Refer to Sinha et al. (2014) for the original SMD problems.

Let $\mathbf{x} = [\hat{x}_{u1}, \hat{x}_{u2}]$ and $\mathbf{z} = [\hat{x}_{l1}, \hat{x}_{l2}]$, $\mathbf{x}, \mathbf{z} \in [0, 1]^d$.

**SMD2**. To bound the output for the given domain, we set

$$F_3 \triangleq -\sum_{i=1}^{r}(x_{u2}^i)^2 - \sum_{i=1}^{r}(x_{u2}^i - \log(0.99 * x_{l2}^i + 0.01))^2,$$

$$f_3 \triangleq \sum_{i=1}^{r} x_{u2}^i - \log(0.99 * x_{l2}^i + 0.01)^2,$$

where $\hat{x}_{u1} \triangleq (x_{u1} + 1)/3$, $\hat{x}_{u2} \triangleq (x_{u2} + 5)/6$, $\hat{x}_{u1} \triangleq (x_{l1} + 1)/3$, and $\hat{x}_{l2} \triangleq x_{l2}/e$.

**SMD6**. The different functions have imbalanced ranges. To balance the different functions in $f$, we set

$$\hat{f}_1 \triangleq f_1/d$$
$$\hat{f}_2 \triangleq f_2/d^2$$
$$\hat{f}_3 \triangleq f_3/d,$$

where $d = 3$, and use $\hat{f} \triangleq \hat{f}_1 + \hat{f}_2 + \hat{f}_3$ as the lower-level objective function. $\hat{x}_b \triangleq (x_b + 1)/3$, for $x_b \in \{x_{u1}, x_{u2}, x_{l1}, x_{l2}\}$.

**SMD12**. To bound the outputs in the domain, we set

$$F_3 \triangleq \sum_{i=1}^{r}(x_{u2}^i - 2)^2 + \sum_{i=1}^{r} \tanh|x_{l2}^i| - \sum_{i=1}^{r}(x_{u2}^i - \tanh x_{l2}^i)^2$$

$$f_3 \triangleq \sum_{i=1}^{r}(x_{u2}^i - \tanh x_{l2}^i)^2$$

We also edited the first upper level constraint to $x_{u2}^i - \tanh x_{l2}^i \geq 1$, $\forall i \in \{1, ..., r\}$, so it becomes an active constraint. One of the lower level constraint was also edited to bound its output range: $x_{l1}^j - \sum_{i=1, i \neq j}^{q}(x_{l1}^i)^3 \geq 0 \ \forall j \in \{1, ..., q\}$. We normalize $\hat{x}_{u1} \triangleq (x_{u1} + 5)/15, \hat{x}_{u2} \triangleq (x_{u2} + 1)/2, \hat{x}_{l1} \triangleq (x_{l1} + 5)/15$, and $\hat{x}_{l2} \triangleq (x_{l2} + \pi/2)/\pi$.

After the following adaptations, we take the mean over input dimensions to ensure that function values do not increase with dimensions. Finally, we normalize the outputs.

## D.4. Energy market

Let $\mathbf{x} \triangleq [\mathbf{x}_1, \mathbf{x}_2]$, where $\mathbf{x}_1$ denotes a price to bid and $\mathbf{x}_2$ denotes a quantity in MW to supply at bid price. $\mathbf{x}_1 \in (0.01, 0.5), \mathbf{x}_2 \in (200, 500)$. We simulate a network with 3 generators that has to fulfill an estimated demand schedule for 3 periods. The generators' parameters are given in Table 2, where $\mathbf{z} \triangleq [\mathbf{z}_1, \mathbf{z}_2]$ are the lower-level variables. $\mathbf{z}_1 \in (0.0, 0.2), \mathbf{z}_2 \in (0.5, 1.5)$. These two variables were selected as a proxy for auxiliary concerns such as efficiency and maintenance costs, on top of operational costs.

The lower-level objective function is denoted as

$$f(\mathbf{x}, \mathbf{z}) \triangleq -\text{cost}(\mathbf{x}, \mathbf{z}) - 2.5 * w_r(\mathbf{z}_1) - 1.5 * w_w(\mathbf{z}_2),$$

where $\text{cost}(\mathbf{x}, \mathbf{z})$ is the operational cost of producing $\mathbf{z}_2$MW of power, simulated by PyPSA. $w_r(\mathbf{z}_1) \triangleq \exp(5 * \mathbf{z}_1) - 1$ and $w_w(\mathbf{z}_2) \triangleq -(\log(-0.75 * \mathbf{z}_2 + 1.15) - (-0.75 * \mathbf{z}_2 + 1.15)) - 0.797$, where $w_r$ and $w_w$ are different nonlinear weighting functions applied to $\mathbf{z}$. If dispatch is not feasible at a point, we set the lower-level objective value with an arbitrary large negative number, and the upper-level objective value at 0.

| type | nominal power | marginal cost | quadratic marginal cost | ramp limit | max p factor |
|------|--------------|---------------|------------------------|-----------|-------------|
| coal | 200 | 0.005 | 0.0005 | $z_1$ | - |
| gas | 100 | 0.015 | 0.0005 | 0.5 | - |
| wind | 60 | 0.02 | 0.005 | - | $z_2$ |

Table 2: Parameters input into PyPSA generator. 'max p factor' refer to 'p_max_pu', the maximum power at a snapshot given as a fraction of nominal power.

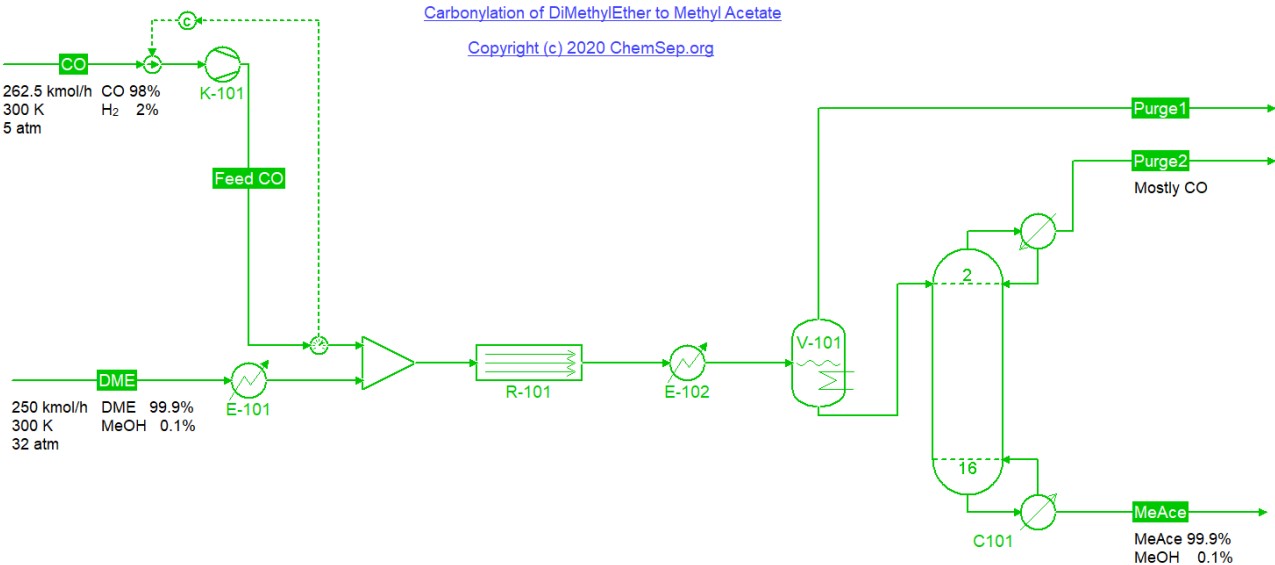

Figure 6: Flowsheet of chemical process. R-101 is the reactor, and C101 is the distillation column.

The upper-level objective function measures profit as

$$F(\mathbf{x}, \mathbf{z}) \triangleq \mathbf{x}_1 * \mathbf{x}_2 * \mathrm{df}(\mathbf{x}_t) - \mathrm{cost}(\mathbf{x}, \mathbf{z}),$$

where $\mathrm{df}(\mathbf{x}_t) \triangleq \min(1, \exp(-10\mathbf{x}_1 + 0.25))$ returns a factor that simulates the demand response of consumers. This implies a disincentive for providers to bid at high prices, because consumers might choose to reduce their electricity usage or look for alternative providers.

We discretized the input space into 15 at each dimension.

### D.5. Chemical process

The flowsheet used is shown in Figure 6, where the output of reactor R101 contains a mix of Methyl Acetate, unreacted DME, and other by-products, and the distillation column C101 separates these products to obtain high purity Methyl Acetate. The flowsheet was adapted from ChemSep, where the recycle streams have been removed to simplify the process. CO and DME are fed in at a fixed flow rate and concentration for all experiments, as indicated in the figure. The distillation feed is always at level 2, and we fixed the output concentration of Methyl Acetate at 99.9%. Note that we can simulate the reactor R101 without the column C101.

The upper- and lower-level parameters to be optimized are defined in Table 3. We discretized the input space into 10 at each dimension, and the variables are normalized to $[0, 1]$.

Let $\mathrm{sim}_{R101}(\mathbf{x}, \mathbf{z})$ be the simulated mass flow of Methyl Acetate (kg/s) at the output of the reactor R101, and $\mathrm{sim}_{C101}(\mathbf{x}, \mathbf{z})$ be the simulated mass flow of Methyl Acetate (kg/s) at the MeAce output of the column C101.

| name | min | max | normalized symbol |
|---|---|---|---|
| Number of levels in distillation column | 5 | 23 | $\mathbf{x}_0 \in [0, 1]$ |
| Temperature of reactor (K) | 455 | 500 | $\mathbf{z}_0 \in [0, 1]$ |
| Number of heating tubes in reactor | 600 | 1500 | $\mathbf{z}_1 \in [0, 1]$ |
| Diameter of heating tubes (m) | 0.02 | 0.065 | $\mathbf{z}_2 \in [0, 1]$ |

Table 3: Parameters of the chemical experiment.

The lower-level objective function is denoted as

$$f(\mathbf{x}, \mathbf{z}) \triangleq \mathrm{sim}_{\mathrm{R101}}(\mathbf{x}, \mathbf{z}) - 1\mathrm{e}\text{-}3 * \mathbf{z}_1^4,$$

where the second term is a penalty on higher temperatures to account for energy costs.

The upper-level objective function is then denoted as

$$f(\mathbf{x}, \mathbf{z}) \triangleq \mathrm{sim}_{\mathrm{C101}}(\mathbf{x}, \mathbf{z}) - 1\mathrm{e}\text{-}4 * \mathbf{x}_0^4,$$

where the second term is a penalty on more levels in the distillation column as it is associated with higher costs. The higher costs could be due to maintenance, energy consumption or equipment cost.

### D.6. Computational resources

The experiments in this paper were done on a computer with AMD Ryzen 7 5700X 8-Core Processor and 64 GB of RAM, unless otherwise specified.

## E. Complexity and efficiency of BILBO

### E.1. Computational complexity of BILBO

For a discretized implementation, the computational complexity of BILBO is affected by the computational complexity of:

- Gaussian processes, $\mathcal{O}(n^3)$,
- Updating trusted sets, $\mathcal{O}(|\mathcal{F}|c)$,
- Selecting function query, $\mathcal{O}(|\mathcal{F}|c)$,
- Optimizing the acquisition function, $\mathcal{O}(c)$,

where $n$ is the number of observations, $c$ is the number of discretized points, and $|\mathcal{F}|$ is the number of blackbox functions.

In a uniform grid discretization, which is used in our implementation, if each dimension is divided into $m$ points, then the cardinality is $c = m^d$, where $d$ is the number of dimensions. Thus, dimensionality $d$ exponentially affects computational complexity when using uniform grid discretization. Adaptive discretization may be able to mitigate the exponential factor of dimensionality, where effective dimension $d_{\mathrm{eff}} \ll d$.

### E.2. Time efficiency of BILBO

While our experiments have shown that BILBO is more sample efficient than nested methods, BILBO does require more computational cost to update trusted sets and select function queries and query points. In the presence of inexpensive lower-level evaluations, BILBO's time efficiency can be lower than that of a nested method. However, do note that BILBO's motivation is for settings with expensive blackbox evaluations, where evaluations can be real-world experiments or simulators that are costly or slow.

In addition, BILBO has advantages in scenarios with noisy observations or multiple lower-level solutions. Nested methods only solve for one solution in each lower-level optimization, and it can be suboptimal in these scenarios. On the other hand, BILBO manages the uncertainty of lower-level estimates in a principled way and allows for multiple lower-level estimates, possibly providing better lower-level estimates to reduce regret more effectively even if each iteration takes more time.

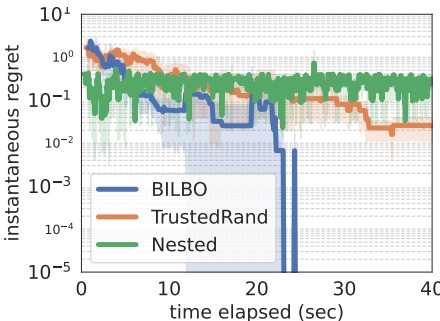

Figure 7: Regret against wall-clock time for the BraninHoo+GoldsteinPrice experiment

In terms of wall-clock time, on a Mac Studio with M2 Ultra over 5 runs and 40 seconds total runtime, for the 2-dimensional BraninHoo+GoldsteinPrice experiment, the average time per BILBO iteration is 0.131s, which is about 1.5 times slower than TrustedRand and about 26 times slower than Nested. Figure 7 shows how regret decreases over wall-clock time. We observed that while the regret for Nested is smaller than BILBO in the initial 5 seconds, Nested's regret quickly plateaus due to suboptimal lower-level estimates of the multimodal lower-level objective. BILBO outperforms Nested subsequently as BILBO converges to a more optimal solution.

