# OpenReview forum: "BILBO: BILevel Bayesian Optimization"
_ICML.cc/2025/Conference — ICML 2025 poster_

### Official Review · Reviewer_kdcy · 2025-02-24

**Overall Recommendation:** 3

**Summary:**

This paper proposes a bi-level Bayesian optimization algorithm for black-box functions. By optimizing both upper- and lower-level problems simultaneously, it improves the sample efficiency. Theoretically, BILBO achieves a sublinear regret bound for common kernels. It also demonstrates strong empirical performance on synthetic and real-world problems.

**Claims And Evidence:**

Yes.

**Essential References Not Discussed:**

No.

**Experimental Designs Or Analyses:**

Yes. I checked the experimental results, and they are mostly sound to me. One issue is that some of the results shown in the experimental section seem to be non-complete, such as the green curves in the Fig. 4 (a, b, c).

**Methods And Evaluation Criteria:**

Yes.

**Other Comments Or Suggestions:**

This approach maintains a set of nearly optimal solutions for the lower-level problem. While discretization is feasible in low-dimensional cases, scaling to even moderately high-dimensional problems is likely challenging.

**Other Strengths And Weaknesses:**

1. This work addresses a significant problem and introduces a novel approach to bilevel Bayesian optimization.
2. Unlike existing methods that requires full lower-level optimization for each upper-level query, the proposed algorithm optimizes both levels concurrently, significantly improving sample efficiency.
3. It also incorporates an infeasibility declaration method and achieves a theoretical sublinear regret bound, extending the classic bound from single-level to bilevel problems.
4. Experimental results demonstrate its effectiveness.

**Questions For Authors:**

What if the lower-level problem is a white-box problem that can be solved via a nonlinear programming solver? Can similar idea work for this case?

**Relation To Broader Scientific Literature:**

Confidence-bound algorithms are extended to bi-level constrained problem, which could be of general interest to Bayesian optimization, bi-level optimization, and multi-armed bandits research.

**Theoretical Claims:**

I checked the key steps for the proof of the main theoretical results, but not all the details. As far as I know, the proofs are correct.

---

> ### Author Rebuttal · Authors · 2025-03-31
>
> Thank you for the detailed review, and for appreciating our novel approach to a significant problem, sample efficiency of our method, sublinear regret bound, and effective experimental results. We would like to clarify the following points.
>
> >One issue is that some of the results shown in the experimental section seem to be non-complete, such as the green curves in the Fig. 4 (a, b, c).
>
> The green curves represent Nested's number of queries over regret. There is an obvious gap at the start of each green curve as the Nested method requires more initial observations of lower-level functions to optimize the lower-level problems separately before estimations begin, compared to BILBO and TrustedRand.
>
> >What if the lower-level problem is a white-box problem that can be solved via a nonlinear programming solver? Can similar idea work for this case?
>
> White-box problems are often inexpensive to compute. We can still apply our algorithm, evaluate the white-box problem at every step and reduce the confidence interval to a point, since there are no noisy observations. If the nonlinear programming solver for the lower-level is preferred (e.g. for computational efficiency), we can also reduce BILBO to a special case where Bayesian optimization is carried out only on the upper-level. However, we would also like to point out that BILBO's advantages are in settings with expensive blackbox evaluations and noisy observations, both of which are not present in this scenario.
>
> We hope these clarifications answer your questions and improve your opinion of our paper.

---

### Official Review · Reviewer_xBiD · 2025-02-25

**Overall Recommendation:** 3

**Summary:**

This paper introduces BILBO (BILevel Bayesian Optimization), a BO (Bayesian Optimization) algorithm used to address constrained bilevel optimization problems where the constraints, the upper objective and the lower objective are all black boxes and assumed expensive to evaluate. BILBO distinguishes itself from the state-of-the-art of derivative-free bilevel BO by its ability to simultaneously optimize the objectives (both upper and lower) and discover the unknown constraints.

The performance of BILBO is studied both theoretically and empirically.

**Claims And Evidence:**

The claims of the paper are the following:

(i) BILBO simultaneously discovers the constraints and optimizes the upper and lower objectives instead of requiring the optimization of the lower objective at each iteration.

(ii) BILBO has a sublinear cumulative regret.

(iii) BILBO performs empirically better than other bilevel optimization methods *in the particular setting considered in the paper*.

Overall, I think the claims are adequately supported. Claim (i) is verified by the definition of BILBO (see Algorithm 1), Claim (ii) is supported by Theorem 4.9 and Claim (iii) is backed up by numerical experiments on low-dimensional problems (see more comments on this particular point below).

**Essential References Not Discussed:**

The very recent work [1] is discussed by the authors but ignored in the numerical experiments. Although comparing BILBO to this solution in numerical experiments would have been a nice addition to the paper, [1] falls under the ICML Concurrent Work policy.

**References**

[1] Ekmekcioglu, O., Aydin, N., & Branke, J. (2024). Bayesian Optimization of Bilevel Problems. arXiv preprint arXiv:2412.18518.

**Experimental Designs Or Analyses:**

The experiments are all low-dimensional (at most 5 dimensions for the SMD synthetic experiments, with 2 and 3 dimensions for the upper and lower objectives, respectively) and the continuous search spaces are discretized with a uniform grid. I believe this is a limitation of the approach, since BILBO requires scanning the whole search space for building trusted sets and solving up to two non-trivial optimization problems at each iteration. I don't think such an expensive process could scale nicely with the search spaces dimensions or with a finer discretization of the search spaces.

To better study the limitations of BILBO, I would have liked to see other numerical results and analyses, such as:
* the immediate regret plotted against wall-clock time for the considered methods,
* the computational complexity of BILBO, especially in terms of the dimensionality of the search spaces and/or the granularity of the discretization,
* the performance of BILBO on bilevel optimization tasks where the lower objective is not expensive, to study how it performs against other bilevel BO solutions that make more restrictive assumptions such as [1, 2].

**References**

[1] Kieffer, E., Danoy, G., Bouvry, P., and Nagih, A. Bayesian optimization approach of general bi-level problems. In Proceedings of the Genetic and Evolutionary Computation Conference Companion, pp. 1614–1621, 2017.

[2] Wang, B., Singh, H. K., and Ray, T. Comparing expected improvement and kriging believer for expensive bilevel optimization. In 2021 IEEE Congress on Evolutionary Computation (CEC), pp. 1635–1642. IEEE, 2021.

**Methods And Evaluation Criteria:**

Because BILBO addresses bilevel optimization in a very particular setting (expensive black-box objectives, expensive black-box constraints), there is no other bilevel optimization method that could directly compete with BILBO (as far as I know). Nevertheless, the paper introduces two benchmarks:

(i) **TrustedRand**: randomly selects queries from trusted sets defined in the paper and observes all the constraints/objectives at once,

(ii) **Nested**: applies BO to the upper objective only, the lower objective is optimized with Sequential Least-Squares Programming (SLQP) at each query.

These two methods are compared against BILBO on synthetic and real-world experiments that make sense for the problem considered in this paper.

**Other Comments Or Suggestions:**

The paper contains many forward pointers (i.e., it refers to Lemmas or Definitions that are introduced later in the paper). I suggest to replace/remove them wherever possible to make reading the paper more pleasant.

**Other Strengths And Weaknesses:**

As far as I understand the paper, BILBO is efficient when the objectives and the constraints are expensive black boxes. I have two comments about this:

* Although this is not a concept introduced by the authors, I think considering expensive black-box constraints is of little interest. In most applications, the constraints are known to the user beforehand and are not expensive to evaluate. Moreover, discovering the constraints during the optimization implies that the constraints are soft (i.e., they can be violated by the optimizer). Including unknown hard constraints (i.e., no measurement can be made if at least one constraint is violated) seems not trivial in your framework.

* Although the sample efficiency of BILBO is definitely one of its strengths, the particular setting within which BILBO is efficient could be seen as a restriction. In fact, as far as I know, most authors justify the use of BO for the upper objective precisely because they require, at each query, an expensive optimization of an inexpensive-to-evaluate lower objective. Given that a significant fraction of bilevel optimization problems occur in this setting, it should be clearly stated in the paper that BILBO can be less efficient (e.g., when comparing the instantaneous regrets w.r.t. the wall-clock times) than other bilevel BO methods when the lower objective is inexpensive to evaluate.

**Questions For Authors:**

Here are some questions (labelled for future reference) to spark the discussion with the authors:

1. Have you studied the performance of BILBO when the trusted spaces are built on discretized versions of continuous search spaces? If yes, what additional assumption do you need to ensure a sublinear regret bound? If no, do you have any intuition about it?

2. How does the immediate regret of BILBO evolves as a function of the wall-clock time?

3. What is the computational complexity of BILBO as a function of the dimensionality of the search spaces? As a function of the cardinality of the uniform grid used for discretization?

4. How would you integrate unknown hard constraints in your framework? How would they impact the performance of BILBO?

5. Do you agree that BILBO could be less efficient than other bilevel BO solutions such as [1, 2] if the lower objective is inexpensive to evaluate? If yes, can you comment on that? If no, can you justify?

**References**

[1] Kieffer, E., Danoy, G., Bouvry, P., and Nagih, A. Bayesian optimization approach of general bi-level problems. In Proceedings of the Genetic and Evolutionary Computation Conference Companion, pp. 1614–1621, 2017.

[2] Wang, B., Singh, H. K., and Ray, T. Comparing expected improvement and kriging believer for expensive bilevel optimization. In 2021 IEEE Congress on Evolutionary Computation (CEC), pp. 1635–1642. IEEE, 2021.

**Relation To Broader Scientific Literature:**

As far as I know, there is no other bilevel optimization method able to discover and optimize the constraints and the objectives when all these functions are expensive black boxes. Although I have some concerns about its scalability, BILBO is useful to solve a bilevel optimization problem under a minimal set of assumptions.

**Theoretical Claims:**

The main theoretical claim is that BILBO has a sublinear cumulative regret. This is backed up by common proof techniques in BO asymptotic analysis and hold with high probability and as long as the search space is compact.

Because the trusted sets are built on discretized search spaces, studying the behavior of BILBO given a discretization would have been a nice addition to the paper.

---

> ### Author Rebuttal · Authors · 2025-03-31
>
> Thank you for your insightful review, and for recognizing our sublinear cumulative regret and sample efficiency when both upper- and lower-level objectives are expensive blackboxes.
> >1. Have you studied the performance of BILBO when the trusted spaces are built on discretized versions of continuous search spaces? If yes, what additional assumption do you need to ensure a sublinear regret bound? If no, do you have any intuition about it?
>
> This is an interesting future research direction that we have not studied. One possible factor on the performance of discretized problems will be the difference between the best discretized point and the optimal point in the continuous domain. This difference may be bounded with respect to the granularity of the discretization, with Lipschitz continuity assumptions. As discretization becomes finer, the performance should converge to that of continuous search spaces.
> >3. What is the computational complexity of BILBO as a function of the dimensionality of the search spaces? As a function of the cardinality of the uniform grid used for discretization?
>
> For a discretized implementation, the computational complexity of BILBO is affected by computational complexity of :
> - Gaussian processes $\mathcal{O}(n^3)$
> - Updating trusted sets $\mathcal{O}(|\mathcal{F}|c)$
> - Selecting function query $\mathcal{O}(|\mathcal{F}|c)$
> - Optimizing the acquisition function $\mathcal{O}(c)$,
>
> where $n$ is the number of observations at an iteration, $c$ is the number of discrete points, and $|\mathcal{F}|$ is the number of blackbox functions. In uniform grid discretization, if each dimension is divided into $m$ points, then the cardinality is $c=m^d$, where $d$ is the number of dimensions. Thus, dimensionality $d$ exponentially affects computational complexity when using uniform grid discretization.
>
> Adaptive discretization may be able to mitigate the exponential factor of dimensionality, where effective dimension $d_\text{eff}\ll d$. However, high dimensionality leads to common challenges that go beyond computational complexity (of BILBO). For example, it is also known to reduce the effectiveness of Gaussian processes as observations become increasingly sparse.
> >4. How would you integrate unknown hard constraints in your framework? How would they impact the performance of BILBO?
>
> We can use variational inference to approximate the Gaussian process posteriors while accounting for infeasible observations, e.g. approximating $p(c(x^*)\mid y(\mathbf{x}_1),c(\mathbf{x}_2)<0)$, where $y(\mathbf{x}_1)$ are noisy feasible observations and $\mathbf{x}_2$ are the observed infeasible points. While $c(\mathbf{x}_2)$ is not observed directly, we can condition on the information that $c(\mathbf{x}_2)<0$ since $\mathbf{x}_2$ is infeasible.
>
> More observations might be required to model constraint boundaries accurately as infeasible observations become less informative. Nevertheless, we can still apply BILBO with the primary adaptation being in the estimation of the posterior belief, while further investigation is needed to assess the effectiveness for hard constraints in practice.
> >5. Do you agree that BILBO could be less efficient than other bilevel BO solutions such as [1, 2] if the lower objective is inexpensive to evaluate? If yes, can you comment on that? If no, can you justify?
>
> The efficiency (in terms of wall-clock time) would depend on the computational cost of each iteration of BILBO versus the cost of evaluating the lower objective, as well as the optimality of lower-level estimates for nested methods such as [1, 2].
>
> While it is possible for BILBO's computational cost to outweigh an inexpensive lower-level objective evaluation, BILBO still has advantages in scenarios with noisy observations or multiple lower-level solutions, which may be more common. Nested methods only solve for one solution for each lower-level optimization, and it can be suboptimal in these scenarios. On the other hand, BILBO manages the uncertainty of lower-level estimates in a principled way and allows for multiple lower-level estimates, possibly providing better lower-level estimates to reduce regret more effectively even if each iteration takes more time.
> >2. How does the immediate regret of BILBO evolves as a function of the wall-clock time?
>
> On a Mac Studio with M2 Ultra, for the 2D BraninHoo+GoldsteinPrice experiment, the average time per BILBO iteration is 0.132s, which is 2x slower than TrustedRand and ~50x slower than Nested.
>
> However, while the regret for Nested decreases faster than BILBO in the initial (~5) seconds, Nested's regret quickly plateaus due to suboptimal lower-level estimates of the multimodal lower-level objective. BILBO outperforms Nested subsequently as BILBO converges to a more optimal solution, supporting our point in the previous question above.
>
> We hope these discussions on the efficiency, complexity and possible extensions of BILBO addressed your questions and will improve your opinion of our work.

---

### Official Review · Reviewer_zVpo · 2025-03-11

**Overall Recommendation:** 3

**Summary:**

The paper proposes a UCB based method for bilevel Bayesian optimization. The proposed method, called BILBO, selects a next point by the upper bound based surrogate model bilevel optimization. The authors provide a regret analysis based on an approach of the well-known GP-UCB analysis.

**Claims And Evidence:**

Since baselines are not well-known methods, efficiency is somewhat difficult to evaluate though I am not sure there does not exist other appropriate existing methods.

**Essential References Not Discussed:**

I don't have any suggestion.

**Experimental Designs Or Analyses:**

The experimental settings are seemingly reasonable.

**Methods And Evaluation Criteria:**

Results on four benchmark functions and two real-word functions are reported. I think the amount of evaluation is sufficient.

**Other Comments Or Suggestions:**

The abbreviation 'LL' is used in some figures, but not defined.

The paper seems only discusses the discrete input setting. It should have been explicitly stated in the main text.

**Other Strengths And Weaknesses:**

S: The topic seemingly has a potential importance, though it has not been widely studied.

W: Significance of performance improvement is somewhat difficult to evaluate because the baseline methods are not popular approach. I guess the Thompson sampling might have been a good baseline.

W: Most of theoretical techniques are directly from well-known Srinivas et al (2010).

**Questions For Authors:**

1) Many lemmas (e.g., Lemma 4.4 and 4.6) should be a probabilistic bound because most of them depend on Corollary 4.2, but is often shown as a deterministic bound. Why? Just omitted?

2) Corollary 4.2 is for simultaneously holding on all possible x, z, h, and t? If so, it should be explicitly written.

3) In the experiments, \sum_h \in F r_h,t is used for the evaluation, but r_F,t can be negative according to (3.5). Is it correct?

4) How is the maximization of (4.6) solved?

**Relation To Broader Scientific Literature:**

Bilevel optimization seemingly often occurs in various scientific fields, and so, I think the topic has a potential importance.

**Theoretical Claims:**

I couldn't fully follow the proof.

---

> ### Author Rebuttal · Authors · 2025-03-31
>
> Thank you for your detailed review and for appreciating the potential of our problem. We would like to address some questions and concerns raised.
>
> >1. Many lemmas (e.g., Lemma 4.4 and 4.6) should be a probabilistic bound because most of them depend on Corollary 4.2, but is often shown as a deterministic bound. Why? Just omitted?
>
> The lemmas are all probabilistic bounds, which were omitted for brevity. We will clarify this in the next version of the paper.
>
> >2. Corollary 4.2 is for simultaneously holding on all possible x, z, h, and t? If so, it should be explicitly written.
>
> Yes, we omitted it for brevity as it was already stated in the directly preceding Definition 4.1. We can add this in the next version of the paper.
>
> >3. In the experiments, \sum_h \in F r_h,t is used for the evaluation, but r_F,t can be negative according to (3.5). Is it correct?
>
> Your interpretation is correct and (3.5) should be $r_F(\mathbf{x}_t, \mathbf{z}_t) ≜ \max(0, F(\mathbf{x}^*, \mathbf{z}^*) - F(\mathbf{x}_t, \mathbf{z}_t))$, which is what we had implemented in our experiments to avoid negating other regret components. Thank you for pointing this out, and we will update (3.5) in the next version of the paper. Lemma C.3. will still hold with minor edits to the proof.
>
> >4. How is the maximization of (4.6) solved?
>
> In our experiments, we discretized the search space so the arg max over all the discrete points can be easily found.
>
> For continuous domains, we can use a continuous representation for the trusted sets, e.g. with hyperrectangles, and use L-BFGS or SLSQP to optimize the acquisition function, similar to TorchBO.
>
> >The paper seems only discusses the discrete input setting. It should have been explicitly stated in the main text.
>
> While we discretized the inputs in our experiments implementation, our theoretical analysis holds for continuous inputs, and it is possible to implement BILBO for continuous inputs in practical settings as discussed above.
>
> >W: Most of theoretical techniques are directly from well-known Srinivas et al (2010).
>
> We would like to clarify that our theoretical techniques are not derived directly from Srinivas et al (2010), but only utilizes an adapted confidence bound in Corollary 4.2. We tackle bilevel challenges that differ from the single-objective, unconstrained optimization in Srinivas et al (2010).
>
> In particular, we introduced a function query selection strategy with conditional reassignment, to select a function query for the decoupled setting, while accounting for the uncertainty of lower-level estimates. The conditional reassignment is also crucial for the exploration of the lower-level objective without repeated lower-level optimizations. These novel components address challenges associated with the lower-level optimization of bilevel problems, and are integral to the derivation of a sublinear regret bound for BILBO.
>
> >The abbreviation 'LL' is used in some figures, but not defined.
>
> 'LL' refers to lower-level. We will define this in the next version of the paper.
>
> We hope our clarifications have addressed your questions and improved your opinion of our paper.

---

> > ### Comment · Reviewer_zVpo · 2025-04-01
> >
> > Thank you for your response.
> >
> > >  our theoretical analysis holds for continuous inputs
> >
> > The theorem depends on the number of candidates in ${\cal X}$ and ${\cal Z}$ (e.g., $\beta_t$ in Theorem 4.9). How does it extend to a continuous space?

---

> > > ### Author Response · Authors · 2025-04-02
> > >
> > > Thank you for your acknowledgment and response.
> > >
> > > >The theorem depends on the number of candidates  in $\mathcal{X}$ and $\mathcal{Z}$ (e.g., $\beta_t$ in Theorem 4.9). How does it extend to a continuous space?
> > >
> > > Indeed, we would need additional assumptions and adjust $\beta_t$ for a continuous space.
> > >
> > > Assume $\forall h \in \mathcal{F}$, $h$ lies in RKHS $\mathcal{H}_{k_h}(D)$ and the RKHS norm is bounded,
> > >
> > > $||h||_{k_h} \leq B_h$,
> > >
> > > where $k_h$ is the corresponding kernel and $D$ is a compact subset of $\mathbb{R}^{d_\mathcal{X} + d_\mathcal{Z}}$. Also, assume that noise sequence $\lbrace \epsilon_{h,t} \rbrace_{t \geq 1}$ is conditionally R-sub-Gaussian. Note that these are common assumptions.
> > > Adapting from [1], we can set $\beta_t$ as $B_h + R \sqrt{2(\gamma_{h,t-1} + 1 + \ln(|\mathcal{F}|/\delta)}$, where $\gamma_{h,t-1}$ is the maximum information gain which can be found in our Appendix B.2.
> > >
> > > The main difference is in the selection of $\beta_t$, while our regret analysis remains valid as Corollary 4.2 holds following Theorem 2 in [1]. We will add this clarification in the next version of the paper.
> > >
> > > (The line breaks in the second sentence were added to ensure proper LaTeX rendering.)
> > >
> > > [1] Chowdhury, S. R., & Gopalan, A. (2017, July). On kernelized multi-armed bandits. In International Conference on Machine Learning (pp. 844-853). PMLR.

---

### Official Review · Reviewer_GE3e · 2025-03-12

**Overall Recommendation:** 3

**Summary:**

This paper introduces BILBO, a Bayesian optimization method for bilevel problems with noisy, constrained black-box objectives. It jointly optimizes both levels using trusted sets from Gaussian process confidence bounds. BILBO provides theoretical regret guarantees and outperforms baselines in empirical evaluations.

**Claims And Evidence:**

The results are theoretical in nature, and proofs are provided. The results are sound.

**Essential References Not Discussed:**

Not aware of any.

**Ethical Review Concerns:**

Not applicable!

**Experimental Designs Or Analyses:**

Not applicable.

**Methods And Evaluation Criteria:**

I believe this is not applicable due to the theoretical nature of the paper.

**Other Comments Or Suggestions:**

Overall, the paper is sound and clear. The technical novelty however seems quite limited as most technical tools are based on standard GP confidence intervals. The paper benefits from a discussion on technical novelty.

**Other Strengths And Weaknesses:**

Strengths:

- Addresses an important, practical problem (efficient bilevel Bayesian optimization).
- Provides clear theoretical guarantees (sublinear regret bound), an important first in this setting.
Trusted set approach effectively reduces expensive lower-level evaluations.
-Good empirical support via both synthetic and real-world experiments.
Weaknesses:

- Limited technical novelty; methods (GP modeling, confidence bounds) are mostly standard.
- Scalability issues due to discretization, especially in higher-dimensional cases.
- Clarity could improve—some definitions and assumptions initially confusing.

**Questions For Authors:**

Could you concisely highlight which specific aspects of BILBO are technically novel, beyond standard Bayesian optimization techniques (GP, confidence bounds)?

How does BILBO practically handle scalability to higher-dimensional problems, given the reliance on discretization?

**Relation To Broader Scientific Literature:**

The paper uses existing methods from Bayesian optimization and extends them to bilevel optimization.

**Theoretical Claims:**

While I did not check all details, the proofs appear generally sound and reasonable.

---

> ### Author Rebuttal · Authors · 2025-03-31
>
> Thank you for your detailed review, and for recognizing the importance and practicality of our bilevel problem, clear and first theoretical guarantees in this setting, and good empirical support. We would like to address the questions raised.
>
> >Could you concisely highlight which specific aspects of BILBO are technically novel, beyond standard Bayesian optimization techniques (GP, confidence bounds)?
>
> BILBO contains the following novel contributions to tackle unique challenges in bilevel Bayesian optimization where the upper-level optimization is constrained by lower-level solutions:
>
> 1. Formulation of the trusted set of optimal lower-level solutions $\mathcal{P}^+_t$ based on lower-level estimates, where we showed that points in the set have lower-level objective regret bound. This reduces the search space and is a key step towards the derivation of regret bound for BILBO.
>
> 2. Function query selection strategy with conditional reassignment of query point. Our function query strategy includes a term for the uncertainty of lower-level estimates, which corresponds to the additional difficulty from lower-level optimization that is not present in standard optimization. We also introduced the conditional reassignment as a way to explore the lower-level objective without the repeated lower-level optimization found in many bilevel Bayesian optimization methods.
>
> >How does BILBO practically handle scalability to higher-dimensional problems, given the reliance on discretization?
>
> One way is via adaptive discretization, which can reduce the effective dimension.
>
> Also, while we had discretized the problems in our experiments, BILBO does not require discretization of continuous inputs. A scalable BILBO implementation for continuous problems could contain a continuous representation of the trusted sets, e.g. with hyperrectangles, and use L-BFGS or SLSQP to optimize the acquisition function, similar to TorchBO.
>
> We hope our clarifications on the novel aspects of our work and scalability discussions addressed your questions and will improve your opinion of our paper.

---

### Decision · Program_Chairs · 2025-05-01

**Decision:**

Accept (poster)

**Comment:**

This paper has largely positive reviews. Reviewers find the setting relevant, the approach new and interesting, and believe the key claims are supported. Reviewers found both the empirical and theoretical contributions valuable, and the writing was described as clear. The main weaknesses raised came down to various technical details involving constraints and how certain discretizations affect performance.

The main issue I am concerned about is the use of search domain discretization in the empirical evaluations, since this can make Bayesian optimization methods perform very badly. Unless I have misunderstood reviews and a quick read of the authors' experiment section, the authors appear to not have evaluated their method with hyperrectangle-based trust sets, which they propose as an alternative for cases where discretization doesn't work. This leaves me worried, but reviewers - some of whom are seasoned experts - thought other aspects of the paper were strong-enough to justify high scores in spite of this issue. Hence, I am going to go with reviewer consensus, and conclude that this weakness, while significant, is not significant enough to warrant outright rejection in light of the work's overall contributions.

In the discussion phase, reviewers suggested the paper should be classified as "weak accept" on balance of the issues. Given the above points, I concur.